# Topography of hippocampal connectivity with sensorimotor cortex revealed by optimizing smoothing kernel and voxel size

Douglas D. Burman[ID]*

Department of Radiology, NorthShore University HealthSystem, Evanston, Illinois, United States of America

* DBurman2@northshore.org

**Data Availability Statement:** All fMRI files are available from the NITRC database, uploaded in 2019: Functional MRI dataset during passive visual stimulation and two motor conditions (sequence learning, paced repetitive tapping)(DOI:10.25790/bml0cm.53).

## Abstract

Studies of the hippocampus use smaller voxel sizes and smoothing kernels than cortical activation studies, typically using a multivoxel seed with specified radius for connectivity analysis. This study identified optimal processing parameters for evaluating hippocampal connectivity with sensorimotor cortex (SMC), comparing effectiveness by varying parameters during both activation and connectivity analysis. Using both 3mm and 4mm isovoxels, smoothing kernels of 0-10mm were evaluated on the amplitude and extent of motor activation and hippocampal connectivity with SMC. Psychophysiological interactions (PPI) identified hippocampal connectivity with SMC during volitional movements, and connectivity effects from multivoxel seeds were compared with alternate methods; a structural seed represented the mean connectivity map from all voxels within a region, whereas a functional seed represented the regional voxel with maximal SMC connectivity. With few exceptions, the same parameters were optimal for activation and connectivity. Larger isovoxels showed larger activation volumes in both SMC and the hippocampus; connectivity volumes from structural seeds were also larger, except from the posterior hippocampus. Regardless of voxel size, the 10mm smoothing kernel generated larger activation and connectivity volumes from structural seeds, as well as larger beta estimates at connectivity maxima; structural seeds also produced larger connectivity volumes than multivoxel seeds. Functional seeds showed lesser effects from voxel size and smoothing kernels. Optimal parameters revealed topography in structural seed connectivity along both the longitudinal axis and mediolateral axis of the hippocampus. These results indicate larger voxels and smoothing kernels can improve sensitivity for detecting both cortical activation and hippocampal connectivity.

## Introduction

Noise can obscure the weak blood oxygen-level dependent (BOLD) response used to detect activation and functional connectivity in functional magnetic resonance imaging (fMRI). A number of approaches have been tried over the years to improve signal detection. Two fundamental considerations are voxel size and the size of the smoothing kernel [1]. Optimal voxel

**Funding:** The author received no specific funding for this work.

**Competing interests:** The author has declared that no competing interests exist.

size should generally be commensurate with activation volume, although due to susceptibility field gradients, smaller voxel sizes may sometimes increase sensitivity despite less total signal intensity [2, 3]. Larger voxel sizes improve signal strength but provide lower spatial resolution, whereas hippocampal studies generally prefer better resolution due its small size; during analysis, voxel dimensions within any plane typically range from 1mm to 3mm [4–12].

Spatial smoothing is used to increase signal-to-noise in BOLD signals and provide smoothness to imaged data [13, 14]. Smoothing expands the size of detected activation by reducing noise; generally, the optimal smoothing kernel is 2–3 times the voxel size, both for individual analysis [15, 16] and for large group analysis, with larger smoothing kernels suggested for group analysis in studies with fewer subjects [17]. Smoothing kernels may degrade the resolution of signal [18], so some studies of the hippocampus avoid smoothing [12], whereas others use a small Gaussian smoothing kernel ranging from 3-6mm full width at half maximum (FWHM) [6, 8, 10, 19, 20]. Nonetheless, several hippocampal studies have used larger smoothing kernels of 8mm [4, 5, 9, 11, 21].

The selection of appropriate processing parameters raises challenges for connectivity studies, especially those designed to examine the influence of hippocampal activity on cortical areas that differ in optimal processing parameters. In functional connectivity studies of the resting state, smoothing has been shown to increase the spatial extent while decreasing the amplitude of the correlated signal [22], at times improving detectability with low-resolution images [23]; these findings emphasize the need to select the appropriate smoothing kernel based on method of seed selection and functional considerations [24]. Generally, the optimal smoothing kernel for connectivity analysis is 2–3 times the voxel size [25], as it is for activation analysis. With few exceptions [26, 27], smoothing kernels used in hippocampal connectivity studies are small, ranging from 2-6mm [28, Heise, 2014 #6998, Menon, 2005 #6997].

The effects of voxel and smoothing kernel size in the hippocampus have not been explored empirically, so their precise relationship to activation and connectivity is unknown. Although effective in activating the hippocampus and its connections with prefrontal cortex (PFC), memory tasks may be less than optimal to identify effects of processing parameters, as the actual size, intensity, and location of memory-related PFC activation may depend on memory content [29–34]. The sensorimotor cortex (SMC), by contrast, shows robust activation based on movements of the represented body part, known from studies of topography [35–38]; furthermore, a recent study showed hippocampal-SMC connectivity restricted to the hand representation during hand movement tasks [39]. As a form of effective connectivity, PPI can identify task-specific influences of hippocampal activity on another region; PPI connectivity from the left hippocampus was observed to be greater than from the right during the sequence learning task, whereas bilateral global analysis was required to reliably detect connectivity during the non-mnemonic repetitive tapping task.

Building on that work, the current study applies PPI to the same dataset to systematically explores the effect of voxel size, smoothing kernel, and seed selection on task-specific activation and connectivity from hippocampus to sensorimotor cortex (SMC). The purpose of the methodological component of this study is not to identify optimal parameters for every case, but to demonstrate that parameters commonly used in hippocampal studies may not be optimal for connectivity studies with cortex. In the current study, an intermediate voxel size (4mm compared to 3mm or 5mm isovoxels) generated a modest but specific increase in cortical activation within the cortical hand representation; 4mm isovoxels size also generated greater connectivity volumes with SMC from all regions except posterior hippocampus (where 3mm isovoxels were advantageous). Larger smoothing kernels consistently increased both cortical activation and connectivity, particularly connectivity from structural seeds. Structural seeds, representing the mean connectivity from all voxels in a specified region of the hippocampus,

proved superior to conventional multivoxel seeds, providing larger connectivity volumes and a demonstrable topographic organization that was otherwise indiscernible.

## Materials and methods

### Subjects

Thirteen normal right-handed adults participated in the study (ages 24–59, mean = 42.3, five females) following written consent; informed consent procedures complied with the Code of Ethics set forth in the Declaration of Helsinki, and were approved by the Institutional Review Board at the NorthShore University HealthSystem / Research Institute.

### Experimental task

Subjects, task, data acquisition, and processing procedures have previously been described in detail [39–41]. Each subject performed a visual/motor task, consisting of 6 cycles of a specified sequence of visual and motor conditions over a period of 6m 4s. This task is illustrated and described in detail elsewhere (see Fig 1 from Burman, 2019).

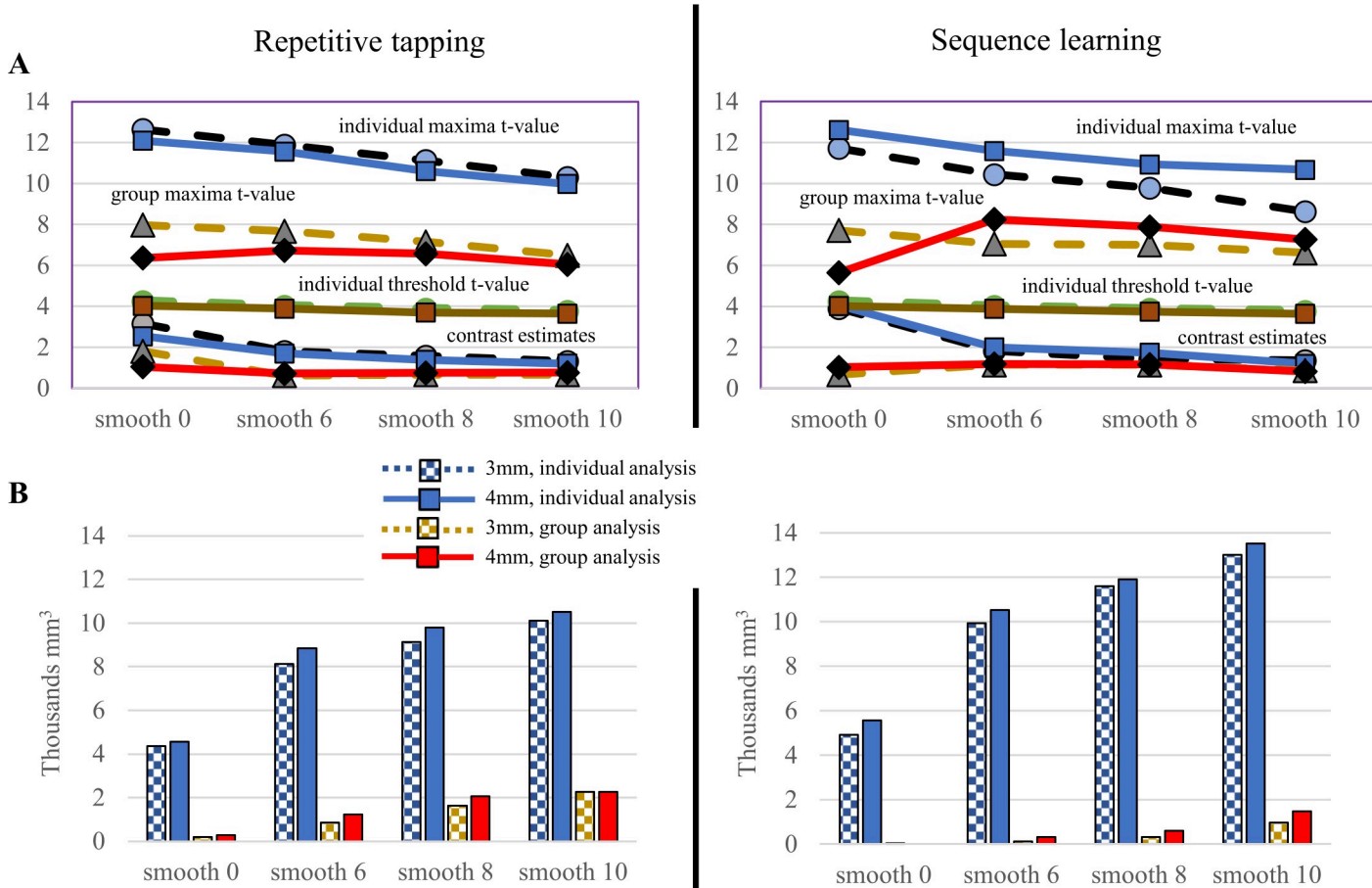

**Fig 1. Effects of smoothing kernel and voxel size on SMC individual and group activation during motor tasks.** (A) Threshold and parameters from the activation maximum, plotted with no smoothing ("smooth 0") and three smoothing kernels applied to 3- and 4-mm isovoxels (solid and dashed lines, respectively). Both for repetitive tapping and sequence learning, parameter differences related to voxel size were small for both individual and group analysis; in both, increases in smoothing kernel above 6mm resulted in small decreases in threshold, magnitude of the contrast, and maximal t-value, with group analyses yielding overall lower values than individual analyses. (B) Despite decreases in parameter values, increases in the size of the smoothing kernel produced larger activation volumes, reflecting the decreased threshold and variability in SMC signal within (blue) and between individuals (gold and red).

Briefly, each cycle included a block of sequential tapping and a block of repetitive tapping, each separated by a passive visual condition. The task ended with 10s of passive fixation on a central cross.

During the sequential tapping block, a 4-button sequence to be remembered was displayed while a metronome ticked at 2 Hz; the subject pressed the remembered sequence of buttons in time with the metronome once the onscreen instructions were replaced with a cross. Subjects repeated the 4-button sequence throughout the 16s block, which ended when a circular checkerboard pattern flickered onscreen for 9s. During this visual block, subjects fixated the center of the pattern and refrained from moving.

During the repetitive tapping block, the subject was instructed to tap the same finger on both hands in synchrony with the metronome. Once the instruction screen was replaced with the number '1', the subject tapped the index finger from both hands; every 4s, the onscreen number increased by one, and the subject changed finger. This motor condition was also followed by the passive visual condition.

Button presses were recorded from the right hand during both motor tasks to verify accurate performance. Behavioral analysis demonstrated motor learning and recall effects during the sequence learning task only; anticipatory movements prior to the metronome demonstrated cognitive control of movements during both repetitive tapping and sequence learning [39].

## MRI data acquisition

Images were acquired using a 12-channel head coil in a 3 Tesla Siemens scanner. Blood-oxygen level dependent (BOLD) functional images were acquired with the echo planar imaging (EPI), using the following parameters: time of echo (TE) = 25 ms, flip angle = 90$^o$, matrix size = 64 x 64, field of view = 22x22 cm, slice thickness = 3 mm, number of slices = 32; time of repetition (TR) = 2000 ms; and the number of repetitions = 182. These parameters produced in-plane voxel dimensions of 3.4 x 3.4mm. A structural T1 weighted 3D image (TR = 1600 ms, TE = 3.46 ms, flip angle = 9$^o$, matrix size = 256 x 256, field of view = 22x22 cm, slice thickness = 1 mm, number of slices = 144) was acquired in the same orientation as the functional images.

## fMRI data processing

An overview of data processing and analysis is shown in S1 Fig.

Data were processed and analyzed using SPM12 software (http://www.fil.ion.ucl.ac.uk/spm). Images were spatially aligned to the first volume to correct for small movements. After removing one cycle from a subject's data set due to excessive head movements, maximal RMS movement for any subject was less than 1.5mm (mean = 0.850 + 0.291mm). Sinc interpolation minimized timing-errors between slices; functional images were coregistered to the anatomical image, normalized to the standard T1 Montreal Neurological Institute (MNI) template, and resliced for two sets of analyses. For one dataset, functional images were resliced during normalization as 3mm isovoxels, potentially improving in-plane image resolution by incorporating differences in signal intensity associated with small head movements. For the other dataset, functional images were resliced during normalization as 4mm isovoxels, potentially increasing sensitivity while decreasing image resolution.

To identify task-related activation, copies of images from each dataset were either left unsmoothed, or smoothed with a 4-, 6-, 8-, 10-, 12-, or 14-mm isotropic Gaussian kernel and filtered with a high pass cutoff frequency of 128s. Evidence for the optimal range of smoothing kernels was sought by extending the range of smoothing typically used during activation and connectivity studies. Conditions of interest were specified for sequence learning, visual, and

repetitive tapping, then modeled for block analysis using a canonical hemodynamic response function.

## Activation analysis

A parameter estimate of the BOLD response to each condition was generated; motor activation was identified for each subject by contrasting mean BOLD responses to motor vs. passive visual conditions. Analyses used an intensity threshold of $p < 0.05$ with a family-wise error (FWE) correction for multiple comparisons, applied to a sensorimotor region of interest (ROI). Typically, this ROI was specified as the overlap between TD and aal atlas labels for post- plus precentral gyrus in the WFUPickatlas toolbox for SPM (http://fmri.wfubmc.edu/software/PickAtlas). An additional analysis used the hand representation as the ROI [39] in order to evaluate whether effects of kernel size and smoothing kernel had a similar effect on activation parameters within the region known to be active.

For each voxel size and smoothing kernel combination, the mean and standard error of activation parameters from all individual subjects were recorded and quantitatively compared; these parameters included t-value threshold (corresponding to that required by the FWE correction), maxima t-values, and contrast amplitudes. Additionally, paired t-tests applied to individual subject data allowed statistical evaluation of different voxel sizes with each smoothing kernel, and of different smoothing kernels with the same voxel size.

For analysis of group effects, BOLD contrasts from individual subjects during the motor memory and repetitive tapping conditions were entered into a 1-sample t-test for each voxel size / smoothing kernel combination, recording the same parameters as for individuals (t-threshold, maxima t-values, and contrast amplitudes). In addition, paired t-tests looked for significant group differences in activation between 6mm and 10mm smoothing kernels.

Activation maps from different conditions were overlapped to allow visual comparisons for differences in the size and location of activation.

## Connectivity analysis

**PPI analysis.** Psychophysiological interactions (PPI) were used to identify task-specific connectivity of the hippocampus with sensorimotor cortex [42–44]. The general approach was described previously [39]. For 4mm isovoxels, 78 voxels were identified from the left hippocampus of the normalized brain and 78 from the right, as delimited by the aal atlas in the WFU PickAtlas toolbox (http://fmri.wfubmc.edu/software/PickAtlas); each seed was wholly contained within the hippocampus. Comparison of connectivity across voxel sizes was limited to the sequence learning task; this preferentially involved the left hippocampus, where 278 voxels were identified from 3mm isovoxels.

At each hippocampal voxel, a contrast was selected and specified to create eigenvariates for all conditions in the statistical model; an interaction term then specified greater effect on activity from the motor condition than from the visual condition. After adjustments for regional differences in timing and baseline activity, a regression analysis showed the magnitude of the BOLD signal that correlated with this interaction term elsewhere in the brain. The magnitude of SMC connectivity was quantified from each hippocampal voxel.

**Seed selection.** Single- and multivoxel approaches were both used to select seeds for connectivity analysis. A 3x3 matrix was created by dividing the hippocampus into thirds of equal distance along the A-P plane, then dividing each section into equal thirds across the medial-lateral plane. Because the head of the hippocampus is enlarged, anterior seeds contained more voxels than middle or posterior seeds, the number of voxels in each structural seed was the same across the medial-lateral plane. Anatomical seeds were labelled by their position within

the matrix (A to C from anterior to posterior, 1 to 3 from medial to lateral) and the sign of connectivity (positive or negative).

A "structural" seed's connectivity map was calculated as the mean connectivity from all voxels within the region; a detailed protocol and SPM12 batch files used to create structural seed connectivity maps have been provided elsewhere [45]. Alternatively, a single "multivoxel" seed was selected from the center of the anatomical location with a 6mm radius; this provided comparable spatial evaluation of connectivity from 3mm isovoxels (a diameter of 4 voxels) and 4mm isovoxels (a diameter of 3 voxels). To minimize asymmetry effects, comparisons in connectivity across voxel sizes were limited to the left hippocampus.

Single-voxel seeds could be based either on anatomical location or function. Anatomical selection was based on spaced intervals within the structural seeds, providing finer-grain analysis that was useful for examining topography. "Functional" seed selection was restricted to voxels within (or adjacent to) a structural seed showing significant connectivity; the single voxel was selected that showed maximal connectivity within the SMC mask. By reflecting individual variability in functional localization, functional seeds sought to provide a better estimate of the magnitude and extent of hippocampal influence on SMC activity. Functional seeds were identified from the left hippocampus for sequence learning ("memseed") and from both the left and right hippocampus for repetitive tapping ("tapseed"), using global analysis from both sides to improve sensitivity [39].

**PPI group analysis.** For the sequence learning task, beta estimates of connectivity from each subject's left hippocampus were entered into a 1-sample random effects analysis; the hand representation delineated from activation analysis was used as the ROI, applying an intensity threshold of $p < 0.05$ with FWE correction. A separate analysis was run for multivoxel and structural seeds at 4 combinations of voxel size (3mm and 4mm) with smoothing kernel (6mm and 10mm).

To improve sensitivity for repetitive tapping, beta estimates of connectivity from each subject's left and right hippocampus were entered into an ANOVA model for global analysis; as before, the hand representation delineated from activation analysis was used as the ROI. This ROI provides optimal sensitivity for detecting effects, as hippocampal connectivity with SMC during volitional finger movements is selective for the hand representation [39].

## Results

### Effects of voxel size and smoothing kernels on SMC activation

Group and individual analyses showed similar trends of voxel size and smoothing kernels on patterns of SMC activation, as described below. During group analyses, differences in group activation between 3mm and 4mm isovoxels appeared minor, as were most smoothing kernel effects *except* for activation volume. At the individual level, however, effects from different voxel sizes and smoothing kernels were both significant.

For both 3mm and 4mm isovoxels, maxima values generally decreased incrementally with larger smoothing kernels. The maximal t-value decreased (Table 1, Fig 1A), a pattern observed during both group and individual analyses; lower maxima t-values were observed during group analysis. Estimates for the magnitude of contrast effects also decreased, especially for individual analyses; the magnitude of effects nearly converged for individual and group analyses for 10mm smoothing (see bottom of graphs). In addition, the threshold to identify significant activation dropped, resulting from less local variability in signal amplitude with increasing kernel size. During both individual and group analyses, the net effect was larger activation volumes with larger smoothing kernels, with 4mm isovoxels always generating larger activation volumes than 3mm isovoxels (Fig 1B).

**Table 1. Effect of voxel size and smoothing kernel on sensorimotor cortical activation from random effects group analysis during motor tasks.**

| Repetitive tapping | Sensorimotor cortex | | | | Hand representation | | |
|---|---|---|---|---|---|---|---|
| *3mm isovoxels* | coordinates | t-value | extent | volume | coordinates | extent | volume |
| smooth 0 | (39,-22,59) | 7.97 | 6 | 162 | (39,-22,59) | 25 | 675 |
| smooth 4 | (33,-16,58) | 7.99 | 11 | 297 | (33,-16,58) | 62 | 1674 |
| smooth 6 | (33,-16,59) | 7.67 | 31 | 837 | (33,-16,59) | 108 | 2916 |
| smooth 8 | (33,-16,59) | 7.15 | 60 | 1620 | (33,-16,59) | 110 | 2970 |
| smooth 10 | (33,-16,56) | 6.49 | 83 | 2241 | (33,-16,56) | 110 | 2970 |
| smooth 12 | (33,-16,56) | 5.88 | 99 | 2673 | (33,-16,56) | 184 | 4968 |
| smooth 14 | (33,-16,56) | 5.53 | 110 | 2970 | (33,-16,56) | 192 | 5184 |
| *4mm isovoxels* | | | | | | | |
| smooth 0 | (42,-16,50) | 6.36 | 4 | 256 | (42,-16,50) | 11 | 704 |
| smooth 4 | (34,-16,62) | 7.11 | 13 | 832 | (34,-16,62) | 40 | 2560 |
| smooth 6 | (34,-16,58) | 6.73 | 19 | 1216 | (34,-16,58) | 47 | 3008 |
| smooth 8 | (34,-16,58) | 6.57 | 32 | 2048 | (34,-16,58) | 50 | 3200 |
| smooth 10 | (34,-16,58) | 6.05 | 35 | 2240 | (34,-16,58) | 53 | 3392 |
| smooth 12 | (34,-16,54) | 5.77 | 42 | 2688 | (34,-16,54) | 53 | 3392 |
| smooth 14 | (34,-16,54) | 5.29 | 44 | 2816 | (34,-16,54) | 53 | 3392 |
| *5mm isovoxels* | | | | | | | |
| smooth 0 | (47,-22,55) | 5.83 | 3 | 375 | (42,-22,55) | 9 | 1125 |
| smooth 4 | (32,-17,60) | 6.37 | 9 | 1125 | (32,-17,60) | 22 | 2750 |
| smooth 6 | (37,-22,55) | 6.13 | 4 | 500 | (37,-22,55) | 9 | 1125 |
| smooth 8 | (42,-22,50) | 6.09 | 17 | 2125 | (42,-22,50) | 37 | 4625 |
| smooth 10 | (42,-22,50) | 5.94 | 23 | 2875 | (42,-22,50) | 41 | 5125 |
| smooth 12 | (37,-22,50) | 5.58 | 27 | 3375 | (37,-22,50) | 45 | 5625 |
| smooth 14 | (37,-22,50) | 4.65 | 10 | 1250 | (37,-22,50) | 36 | 4500 |
| **Sequence learning** | | | | | | | |
| *3mm isovoxels* | coordinates | t-value | extent | volume | coordinates | extent | volume |
| smooth 0 | (-36,-13,59) | 7.71 | 1 | 27 | (-36,-13,59) | 1 | 27 |
| smooth 4 | (-42,-37,55) | 7.16 | 1 | 27 | (-45,-37,55) | 15 | 405 |
| smooth 6 | (-42,-37,53) | 7.05 | 4 | 108 | (-39,-40,47) | 61 | 1647 |
| smooth 8 | (-42,-37,53) | 6.79 | 12 | 324 | (-37,-37,47) | 62 | 1674 |
| smooth 10 | (-39,-37,50) | 6.62 | 35 | 945 | (-37,-37,47) | 62 | 1674 |
| smooth 12 | (-39,-34,53) | 5.67 | 48 | 1296 | (-39,-37,50) | 243 | 6561 |
| smooth 14 | (-42,-34,53) | 5.23 | 57 | 1539 | (-42,-34,53) | 276 | 7452 |
| *4mm isovoxels* | | | | | | | |
| smooth 0 | -- | -- | 0 | 0 | (-38,-16,54) | 3 | 192 |
| smooth 4 | (-42,-36,54) | 6.71 | 3 | 192 | (-42,-36,54) | 17 | 1088 |
| smooth 6 | (-42,-36,54) | 7.07 | 5 | 320 | (-42,-36,54) | 60 | 3840 |
| smooth 8 | (-42,-36,54) | 6.76 | 11 | 704 | (-42,-36,54) | 77 | 4928 |
| smooth 10 | (-38,-36,50) | 6.23 | 27 | 1728 | (-38,-36,50) | 80 | 5120 |
| smooth 12 | (-38,-36,50) | 5.86 | 32 | 2048 | (-38,-36,50) | 80 | 5120 |
| smooth 14 | (-38,-36,50) | 5.22 | 37 | 2368 | (-38,-36,50) | 81 | 5184 |
| *5mm isovoxels* | | | | | | | |
| smooth 0 | (-43,-22,55) | 5.73 | 1 | 125 | (-43,-22,55) | 2 | 250 |
| smooth 4 | (-43,-37,55) | 6.79 | 2 | 250 | (-43,-37,55) | 4 | 500 |
| smooth 6 | (-43,-22,55) | 5.45 | 1 | 125 | (-43,-22,55) | 2 | 250 |
| smooth 8 | (-38,-37,50) | 6.86 | 6 | 750 | (-38,-37,50) | 40 | 5000 |
| smooth 10 | (-38,-37,50) | 6.53 | 12 | 1500 | (-38,-37,50) | 46 | 5750 |

(*Continued*)

**Table 1.** (Continued)

| Repetitive tapping | Sensorimotor cortex | | | | Hand representation | | |
|---|---|---|---|---|---|---|---|
| smooth 12 | (-38,-37,50) | 5.89 | 18 | 2250 | (-38,-37,50) | 54 | 6750 |
| smooth 14 | -- | -- | -- | -- | -- | -- | -- |

Region-of-interest (ROI) activation met an intensity threshold of p = 0.05 with a family-wise error (FWE) correction. The sensorimotor ROI represented the pre/postcentral gyrus overlap between the aal and talairach atlases, as designated in the WFU PickAtlas toolbox for SPM; the hand representation was that used in a previous report (Burman, 2019). Coordinates and t-values represent activation maxima.

Larger smoothing kernels thus provided greater sensitivity for detecting activation by reducing noise (random variability in signal). Comparing changes in activation volume with different ROIs provides further insight (Table 1). With the SMC mask, large increases in activation volume were observed as the smoothing kernel increased 6mm to 10mm, volume increases of 84.2–440.0%. These increases were much smaller within the hand representation, never exceeding 33.3% and nearly non-existent for 3mm isovoxels (volume increases of 1.9–1.6% for repetitive tapping and sequence learning, respectively). This suggests larger smoothing kernels are particularly effective for improving statistical detection by reducing signal variability outside the activated region, thus requiring a smaller mean difference in signal to detect activation.

Because data from the same individuals were processed with different voxel size and smoothing kernels, differences in activation parameters resulting from different processing methods could not be measured directly within SPM for each individual. The standard error for mean differences in parameters across the group of subjects often differed greatly for both repetitive tapping and sequence learning; nonetheless, paired t-tests showed that individuals consistently had lower activation thresholds and greater activation volumes with larger (4mm) isovoxels, regardless of smoothing kernel (Table 2).

For each task, the effects of different smoothing kernels on activation parameters were also evaluated with paired t-tests during each motor task (Tables 3 and 4). With 3mm isovoxels, larger smoothing kernels during both motor tasks generated higher activation volumes, but significantly lower maxima t-values, contrast magnitudes, and activation thresholds. Often, these patterns were also observed with 4mm isovoxels; however, no differences were observed between 8mm and 10mm smoothing kernels for activation threshold during repetitive tapping (Table 3), or for the maxima t-values and contrast magnitudes during sequence learning (Table 4).

To better understand the underlying effects of parameter changes, a subject's beta estimates of the contrast magnitude were plotted for a series of voxels adjacent to the maximum in three planes. This subject was selected because SMC activation before smoothing was most representative of the group; the volume of activation was the median (2970mm$^3$ with 3mm isovoxels, compared with a mean volume of 2985mm$^3$ for the group) and the activation maximum was at the median location, near the mean along each of 3 axes (-36,-28,53 compared to the mean of -36,-28,54). Activation during the sequence learning task was recorded as a function of motor activity embedded within noise; the contrast beta estimate at each voxel reflected the magnitude of activation, with the standard error applied to identify activation above the mean noise level (beta estimate of 0).

Regardless of voxel size, activation of 8-12mm width was observed before smoothing along the x-axis, (Fig 2, left); activation extended further in the y-axis, as it included both pre-and postcentral gyrus. Surrounding this peak of activation was a trough of inactivity, surrounded

**Table 2. Effects of voxel size on SMC activation.**

| maxima value | Repetitive tapping | | | |
|---|---|---|---|---|
| | 0mm | 6mm | 8mm | 10mm |
| **mean difference**[†] | -.549±.371 | -.322±.495 | -.523±.238 | -.319±.263 |
| p-value | .165 | .529 | .048* | .250 |
| **contrast magnitude** | | | | |
| mean difference | -.618±.322 | -.106±.105 | -.192±.115 | -.116±.095 |
| p-value | .079 | .334 | .121 | .245 |
| **threshold t-value** | | | | |
| mean difference | -.217±.001 | -.124±.001 | -.171±.050 | -.127±.003 |
| p-value | < .001* | < .001* | < .001* | < .001* |
| **activation volume** | | | | |
| mean difference | 213.1±302.1 | 732.5±196.8 | 642.9±275.6 | 421.1±306.7 |
| p-value | .002* | .003* | .003* | .004* |
| **maxima value** | Sequence learning | | | |
| | 0mm | 6mm | 8mm | 10mm |
| **mean difference** | -.908±.342 | -.509±.347 | -.410±.193 | .517±.339 |
| p-value | .021* | .168 | .056 | .154 |
| **contrast magnitude** | | | | |
| mean difference | .143±.381 | .169±.346 | .314±.233 | -.165±.165 |
| p-value | .714 | .635 | .204 | .338 |
| **threshold t-value** | | | | |
| mean difference | -.217±.001 | -.124±.001 | -.122±.002 | -.127±.003 |
| p-value | < .001* | < .001* | < .001* | < .001* |
| **activation volume** | | | | |
| mean difference | 635.3±192.8 | 609.2±353.6 | 327.2±450.4 | 532.0±507.4 |
| p-value | .001* | .001* | < .001* | < .001* |

[†] At each smoothing kernel, differences in value were calculated for each individual from subject data resliced with 3mm vs. 4mm isovoxels; the mean difference ± standard error is listed.

* p<0.05 in a paired t-test.

by an annular region with slightly elevated activity. With 6mm smoothing, peak activation was reduced in amplitude, but the trough of inactivity was lost. The surrounding cortex showed flat activity slightly above the mean level of noise; nonetheless, the functional width of 10mm peak activation was maintained. For 5mm isovoxels, peak activation began to flatten with 6mm smoothing, becoming indistinguishable from the elevated baseline activity with 10mm smoothing. Activation was indistinguishable for 3mm and 4mm isovoxels at 10mm smoothing, with the area of peak activation broadened slightly.

Across a broad range of smoothing kernels, 4mm isovoxels appeared optimal. The volume of activation increased incrementally with the size of smoothing kernel until asymptotic levels were reached at 10mm smoothing, a pattern observed for both sequence learning and repetitive tapping (S2 Fig). By contrast, activation volumes were low for 3mm and 5mm isovoxels with small smoothing kernels, jumping to high volumes when the activation curve flattened and became non-localized.

To summarize, 4mm isovoxels provided greater sensitivity to cortical activation for the group than 3mm isovoxels (lower threshold and higher activation volume), as did 10mm compared with 6mm smoothing kernels (for both 3mm and 4mm isovoxels). Voxel-wise analysis from an individual subject supports this interpretation, demonstrating graded but localized

**Table 3. Smoothing kernel effects on SMC activation during repetitive tapping.**

| maxima t-value | 3mm isovoxels | | | |
|---|---|---|---|---|
| | 0mm | 6mm | 8mm | 10mm |
| *group mean* | *12.64±1.34* | *11.89±1.36* | *11.13±1.35* | *10.30±1.27* |
| 0mm | NA | .075 | .004* | < .001* |
| 6mm | -- | NA | < .001* | < .001* |
| 8mm | -- | -- | NA | < .001* |
| **contrast magnitude** | | | | |
| *group mean* | *3.16±.34* | *1.80±.21* | *1.57±.23* | *1.31±.19* |
| 0mm | NA | < .001* | < .001* | < .001* |
| 6mm | -- | NA | < .001* | < .001* |
| 8mm | -- | -- | NA | < .001* |
| **threshold t-value** | | | | |
| *group mean* | *4.24±.01* | *4.01±.02* | *3.87±.02* | *3.77±.02* |
| 0mm | NA | < .001* | < .001* | < .001* |
| 6mm | -- | NA | < .001* | < .001* |
| 8mm | -- | -- | NA | < .001* |
| **activation volume** | | | | |
| *group mean* | *4340.8±1023.6* | *8129.1±2093.1* | *9134.3±2442.8* | *10089.7±2754* |
| 0mm | NA | .005* | .006* | .007* |
| 6mm | -- | NA | .017* | .018* |
| 8mm | -- | -- | NA | .035* |
| **maxima t-value** | **4mm isovoxels** | | | |
| | 0mm | 6mm | 8mm | 10mm |
| *group mean* | *12.09±1.26* | *11.57±1.32* | *10.61±1.26* | *9.98+1.23* |
| 0mm | NA | .366 | .005* | .001* |
| 6mm | -- | NA | .002* | < .001* |
| 8mm | -- | -- | NA | < .001* |
| **contrast magnitude** | | | | |
| *group mean* | *2.54±.21* | *1.70±.19* | *1.38±.20* | *1.20±.20* |
| 0mm | NA | .006* | .001* | < .001* |
| 6mm | -- | NA | .031* | < .001* |
| 8mm | -- | -- | NA | .237 |
| **threshold t-value** | | | | |
| *group mean* | *4.03±.01* | *3.89±.02* | *3.70±.05* | *3.64±.02* |
| 0mm | NA | < .001* | < .001* | < .001* |
| 6mm | -- | NA | .002* | < .001* |
| 8mm | -- | -- | NA | .271 |
| **activation volume** | | | | |
| *group mean* | *4553.8±1061.1* | *8861.5±2197.5* | *9777.2±2541.7* | *10510.8±2839* |
| 0mm | NA | .004* | .006* | .008* |
| 6mm | -- | NA | .034* | .031* |
| 8mm | -- | -- | NA | .045* |

* $p < 0.05$ in a paired t-test.

responses within the hand representation that reached its maximal volume of activation with a 10mm smoothing kernel. This 10mm smoothing kernel matched the functional width of activation before smoothing.

**Table 4. Smoothing kernel effects on SMC activation during sequence learning.**

| maxima t-value | 3mm isovoxels | | | |
|---|---|---|---|---|
| | 0mm | 6mm | 8mm | 10mm |
| *group mean*[†] | *11.72±.55* | *10.44±.77* | *9.78±.79* | *8.63±.68* |
| 0mm | NA | .055 | .012* | < .001* |
| 6mm | -- | NA | .067 | < .001* |
| 8mm | -- | -- | NA | .001* |
| **contrast magnitude** | | | | |
| *group mean* | *3.91±.45* | *1.84±.24* | *1.43±.15* | *1.37±.19* |
| 0mm | NA | < .001* | < .001* | < .001* |
| 6mm | -- | NA | .075 | .014* |
| 8mm | -- | -- | NA | .630 |
| **threshold t-value** | | | | |
| *group mean* | *4.24±.01* | *4.01±.02* | *3.87±.02* | *3.78±.02* |
| 0mm | NA | < .001* | < .001* | < .001* |
| 6mm | -- | NA | < .001* | < .001* |
| 8mm | -- | -- | NA | < .001* |
| **activation volume** | | | | |
| *group mean* | *4932.7±983.7* | *9936.0±2070.7* | *11576.8±2354* | *13001.5±2607* |
| 0mm | NA | .001* | < .001* | .001* |
| 6mm | -- | NA | < .001* | < .001* |
| 8mm | -- | -- | NA | < .001* |
| **maxima t-value** | **4mm isovoxels** | | | |
| | 0mm | 6mm | 8mm | 10mm |
| *group mean* | *10.81±.39* | *9.93±.71* | *9.37±.72* | *9.1±.71* |
| 0mm | NA | .133 | .021* | .009* |
| 6mm | -- | NA | .002* | .027* |
| 8mm | -- | -- | NA | .430 |
| **contrast magnitude** | | | | |
| *group mean* | *4.06±.51* | *2.01±.40* | *1.74±.36* | *1.20±.15* |
| 0mm | NA | < .001* | < .001* | < .001* |
| 6mm | -- | NA | .104 | .074 |
| 8mm | -- | -- | NA | .138 |
| **threshold t-value** | | | | |
| *group mean* | *4.03±.01* | *3.89 ±.02* | *3.75±.02* | *3.64±.02* |
| 0mm | NA | < .001* | < .001* | < .001* |
| 6mm | -- | NA | < .001* | < .001* |
| 8mm | -- | -- | NA | < .001* |
| **activation volume** | + | | | |
| *group mean* | *5568.0±1008.0* | *10545.2±2046* | *11904.0±2361* | *13533.5+2855* |
| 0mm | NA | .001* | < .001* | .002* |
| 6mm | -- | NA | .001* | .006* |
| 8mm | -- | -- | NA | .021* |

* p<0.05 in a paired t-test.

## Effect of smoothing kernels on hippocampal motor activation

Group analysis revealed no motor activation within the hippocampus, regardless of voxel size or smoothing kernel.

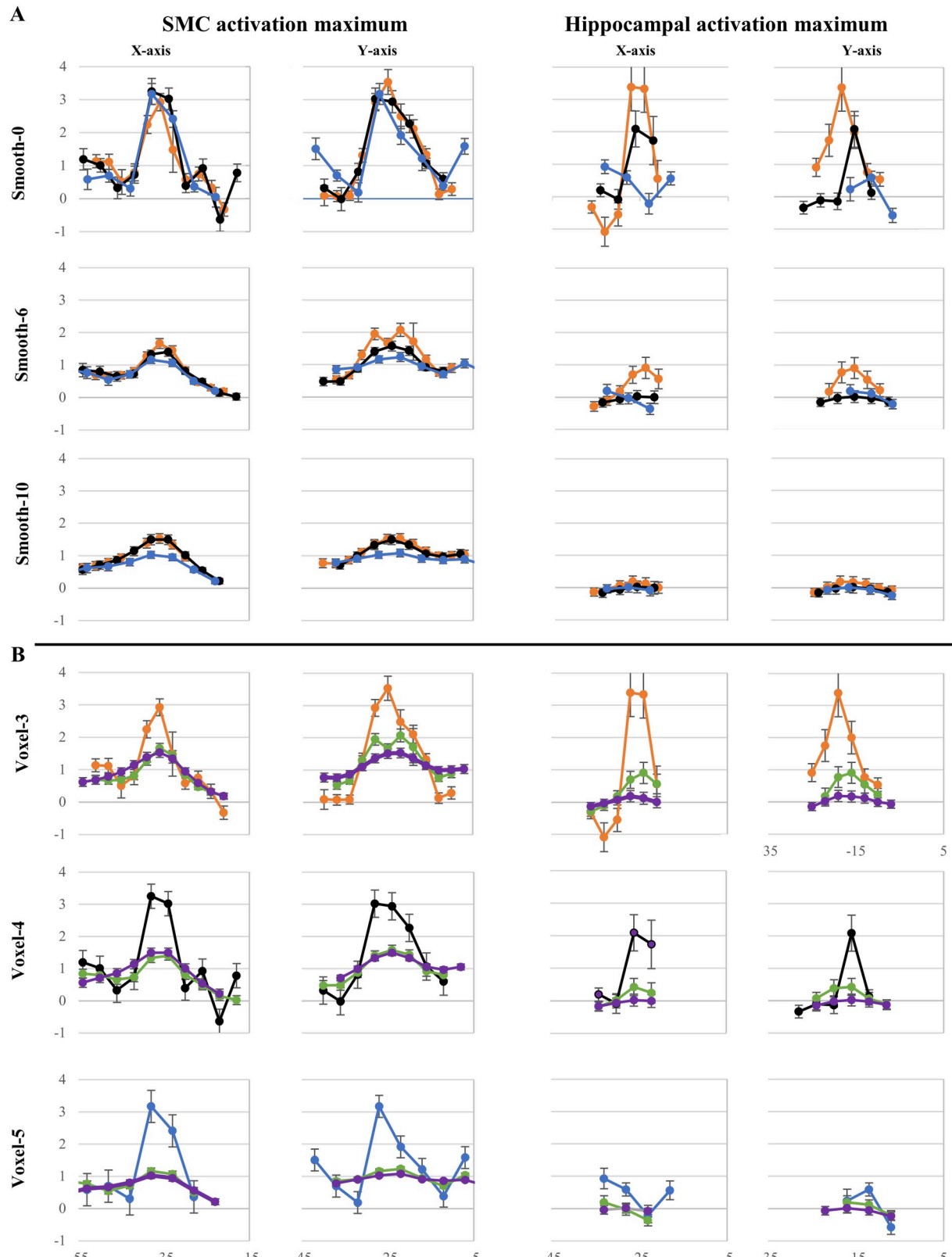

**Fig 2. Effects of smoothing and voxel size on measures of activation in SMC and the hippocampus from a single subject.** The beta parameter estimate from the sequence learning contrast was plotted for each voxel along the x- and y-axis of the activation maximum. Larger

smoothing kernels invariably reduced the peak amplitude, slightly elevating baseline levels in SMC but not the hippocampus. Smoothing effects interacted with voxel size, with peak activation degraded most by large smoothing kernels in 5mm voxels. Orange = 3mm isovoxels, black = 4mm isovoxels, blue = 5mm isovoxels; ordinate values represent beta estimates of the contrast magnitude; error bars represent standard errors.

The effect of smoothing on hippocampal activity was nonetheless examined in the representative subject; by eliminating the statistical threshold, activation during the sequence learning task was observed as a function of hippocampal motor activity embedded within noise. Again, the contrast beta estimate at each voxel represented the magnitude of activation, with the standard error applied to identify activation above the mean noise level (beta estimate of 0).

With 3mm voxels and no smoothing, the activation maxima in this subject lay medial, non-adjacent to the edge (Fig 2, right). Activation was elevated for 3–4 voxels, decreasing from the maxima in every direction; this drop in activity level turned to deactivation near the lateral border. Activation with 4mm isovoxels was similarly positioned although less intense, consisting of 2 voxels in width, but no de-activation was observed. Activation with 5mm isovoxels was bifurcated and offset, greatly reduced in magnitude.

With 6mm smoothing, activation was present for 3mm isovoxels but reduced in amplitude, whereas de-activation was no longer observed. Neither activation nor deactivation were observed following smoothing for 4mm or 5mm isovoxels.

Individual variability in activation surrounding the hippocampal maximum is illustrated in S3 Fig. Before smoothing, the area of activation surrounding the peak extends 12-16mm in diameter for both 3mm and 4mm isovoxels. Activation was reduced or eliminated on both tasks when smoothing 4mm isovoxels, but also when smoothing 3mm isovoxels during sequence learning. The effect of smoothing on mean activation magnitude depended on the similarity of response across adjacent voxels; it had little effect at those locations where nearby voxels showed similar activity, yet when the activity of neighboring voxels differed, responses at individual voxels could change markedly by smoothing.

To summarize, hippocampal activation was not observed during group analysis, but individual analysis revealed the best detection of activation / deactivation without smoothing; in some instances, activation could improve using 3mm voxels.

### Hippocampal connectivity: Structural seeds

**Effects of voxel size and smoothing on temporal patterns of activity in the hippocampus.** To better understand the underlying basis for parameter effects on connectivity, a detailed analysis of these effects was undertaken on the representative subject used for Fig 2.

PPI connectivity is based on moment-by-moment fluctuations in the fMRI signal involved in task performance, captured by the eigenvariate time series ("waveform") from the seed region. Fig 3 examines the eigenvariate from the hippocampal activation minimum and maximum, comparing waveforms across three voxel sizes and three smoothing kernels. At the activation minimum, waveform differences between different voxel sizes appeared small when evaluated with the same smoothing kernel (Fig 3A, left); the waveform for all voxels was reduced in amplitude with larger smoothing kernels. At the activation maximum, by contrast, large difference in the amplitude of the waveform were observed before smoothing, as 5mm voxels barely resemble the largest-amplitude waves of the 3mm voxels (Fig 3A, right). Voxel-related differences, as well as the amplitude of all waveforms, diminished with larger smoothing kernels. Fig 3B shows smoothing effects. Although the effect was greater at the activation maximum, 6mm (green) and 10mm smoothing kernels (purple) progressively decreased the amplitude of the waveform.

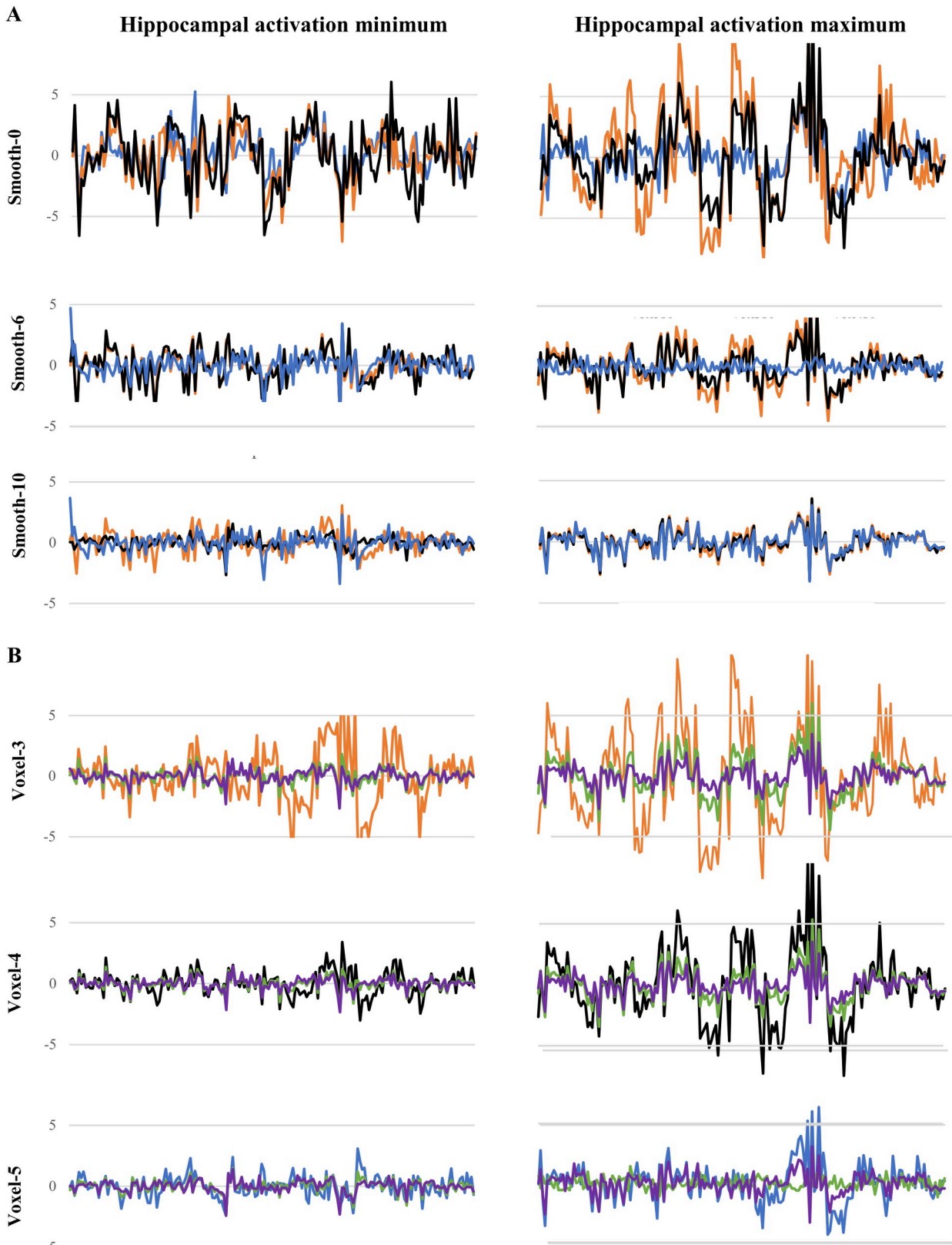

**Fig 3. Effects of smoothing and voxel size on the temporal waveform of task activity in the hippocampus.** (A) Effects of voxel size on temporal waveforms of signal intensity across the duration of the task, plotted separately for different smoothing kernels. At the activation maximum, unlike

the minimum, 3mm isovoxels generated the largest amplitude and 5mm isovoxels the smallest-amplitude waveforms; these differences were reduced with 6mm- and eliminated with 10mm-smoothing. Data color-coded as in Fig 2. (B) Effects of smoothing kernel on temporal waveforms, plotted separately for different voxel sizes. Reductions in waveform amplitude are apparent with a 6mm- (green) and 10mm-smoothing kernel (purple). Waveforms without smoothing are color-coded as above.

While increases in voxel size and smoothing kernel reduced the amplitude of the eigenvariate waveform, they also altered the waveform itself. In Fig 4A, waveforms from all voxels surrounding the maxima (top) for sequence learning show considerable diversity from the waveform at the activation maximum itself (black), sometimes reversing polarity. The mean waveform of neighboring voxels shows similarities, but also differences from the maxima waveform (middle); the same applies to the waveform at the minima voxel compared to its neighbors. The waveform at the minima and maxima voxel generally mirrored each other, yet only weakly reflected the timing of the sequence learning task. Interestingly, the maxima and minima shared a neighboring voxel (see arrow), reflecting the rapid change in response properties within the hippocampus. In this study, the local activation signal within the hippocampus was not useful for guiding PPI seed placement.

Detailed connectivity analysis was carried out on this subject's functional seed, a single voxel generating maximal connectivity shown to generate significant connectivity during group analysis [39]. The eigenvariate waveform of hippocampal activity is used to generate a "predicted" PPI waveform used to evaluate cortical connectivity from comparison with its actual activity. The predicted waveform and SMC activity at the PPI maxima are shown in Fig 4B, both before and after smoothing with a 10mm kernel (applied to 4mm voxels). Estimates of the magnitude of the matching activity relative to baseline noise are shown below for nearby SMC voxels in the same x-axis. Before smoothing, three adjacent voxels showed elevated connectivity, with high variability in SMC activity; the predicted waveform of nearby voxels varied in amplitude (bottom) to provide the best match between the actual activity and the predicted waveform. After smoothing, there was little variability in the predicted waveform or SMC activity along the x-axis, demonstrating more widespread connectivity despite a smaller amplitude in the predicted waveform. The predicted waveform showed an increase in activity associated with the sequence learning task, although its onset was variable.

**Effects of voxel size and smoothing on hippocampal connectivity.** Inverse connectivity from many regions of the hippocampus was observed during sequence learning (Fig 5 and Table 5). For a given combination of voxel size and smoothing kernel, a structural seed generated a larger volume of SMC connectivity than the corresponding multivoxel seed, even when the parameter estimate at the postcentral or precentral maximum was larger for the multivoxel seed. In some cases, differences in methodology could determine whether connectivity remained significant following an additional correction for testing these nine regions (stars). The connectivity from A1 and A2 seeds, for example, were significant only for structural seeds derived from 4mm isovoxels with 10mm smoothing.

In anterior (A1-A3) and middle hippocampus (B1-B2), the volume of connectivity during random effects analysis was typically greater for 4mm isovoxels, both for structural and multivoxel seeds; only in posterior hippocampus (C1-C3) was the volume of connectivity greater for 3mm isovoxels (see also Table 6). In paired t-tests, beta parameter estimates in structural seeds were significantly larger for 4mm voxels and also with the larger smoothing kernel (10mm), especially at postcentral maxima (Table 6). Overall, the volume of connectivity was larger for structural seeds than multivoxel seeds (invariably so when effects survived the FWE correction), even when maxima parameter estimates were significantly smaller (observed only with the smaller 6mm smoothing kernel).

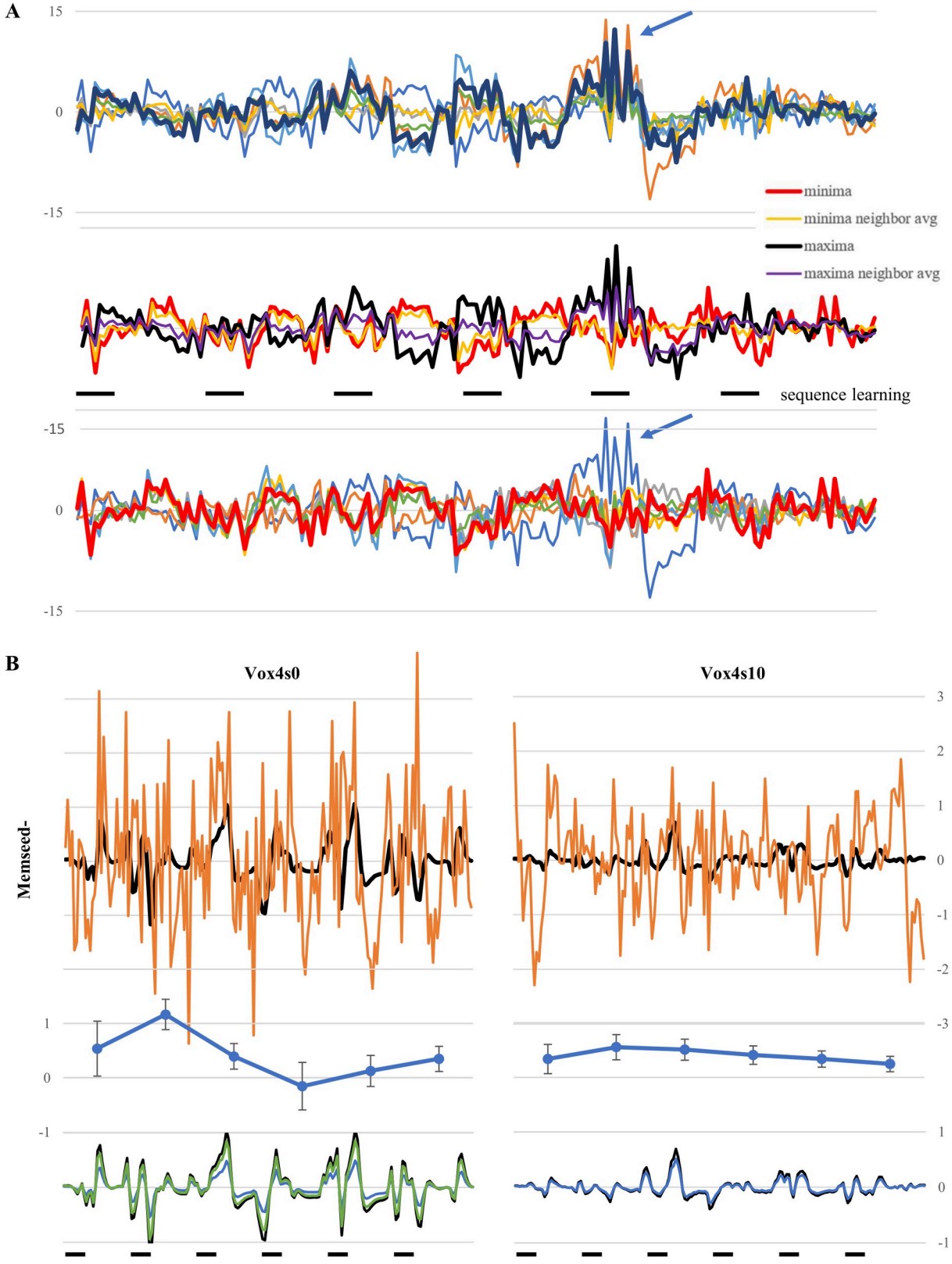

**Fig 4. Temporal waveforms in the hippocampus and SMC and their role in connectivity analysis.** (A) Eigenvariate waveforms for the duration of the task are shown for the activation maxima and neighboring voxels (top), the activation minima and neighboring voxels

(bottom), and the mean of the neighboring voxels in relationship to these (middle). Considerable variability in the pattern of activity is seen in neighboring voxels, with one voxel neighboring both maxima and minima (arrow), whereas overall activity patterns were weakly related to the timing of sequence learning. (B) The PPI waveform scaled for best fit to cortical activity at the memseed⁻ maxima is shown before and after smoothing with a 10mm smoothing kernel, along with SMC activity fit to the data (top). The amplitude of the best-fit waveform at the maxima was larger before smoothing but accompanied by high variability in SMC activity, so fewer voxels along the x-axis fit the data better than chance (middle). Voxels two-removed from the maxima also showed more variability in the amplitude of the predicted waveform before smoothing (bottom).

Connectivity demonstrated through different methodologies overlapped extensively, particularly when the volume of connectivity was similar (see Fig 5, seed B1- at bottom left). When one method generated a much larger volume, however, an additional locus of connectivity could appear (in bottom right of Fig 5, see precentral connectivity for S3 method at seed C1⁻). In such a case, the volume of connectivity reflected the sensitivity of the method.

To summarize, structural seeds improved volume and sensitivity for detecting connectivity compared to traditional multivoxel seeds. For structural seeds, larger voxels (4mm vs. 3mm) improved connectivity measures except in posterior hippocampus; a larger smoothing kernel (10mm vs. 6mm) improved connectivity measures for both 3mm and 4mm voxels.

**Effects of smoothing on connectivity during repetitive tapping.** For repetitive tapping, connectivity was examined using global analysis from both the left and right hippocampus, which improves sensitivity for detecting connectivity [39]. We further examined smoothing kernel effects on connectivity from structural seeds during repetitive tapping, comparing effects of 6mm and 10mm smoothing kernels on 4mm isovoxels.

Structural seeds showed similar maxima locations in pre- and postcentral gyri with both smoothing kernels, but the extent of connectivity was invariably greater with the larger kernel (Table 7). Depending on the size of the smoothing kernel, a topographic arrangement of connectivity was discernible (Fig 6). In the central third of the hippocampus, a topography was observed along the medial / lateral axis with the 6mm smoothing kernel (Fig 6A, left); connectivity from more lateral seeds (cyan and navy blue) extended progressively further posterior and superior. This topography was largely obscured with 10mm smoothing (Fig 6A, right), because larger clusters of connectivity from the middle seed (cyan) produced extensive overlap.

Along the anterior / posterior axis of the hippocampus, however, the topographic organization was more apparent with the 10mm smoothing kernel (Fig 6B). Connectivity for each seed was centered in the precentral gyrus, barely extending across the central sulcus into the postcentral gyrus. From the anterior hippocampal seed A1 (red), intense connectivity projected further anterolateral; the middle seed B1 projected superomedial (yellow), whereas the posterior seed C1 projected further medial and posterior (green). Because its clusters were smaller and spotty, this topography was indiscernible with the 6mm smoothing kernel.

Topography was also demonstrable from individual voxels with 10mm smoothing along both the medial-lateral and longitudinal axes (Fig 7). In the medial-lateral plane, the organization described above is seen in the pattern of overlapping connectivity maps (Fig 7A, left), as well as differential connectivity maps that illustrate which seeds generated the greatest amplitude of connectivity (Fig 7A, right). Along the longitudinal axis, less overlap was observed from individual voxels (Fig 7B), perhaps because of the greater distance between sampled voxels along this axis. From voxels in both planes, connectivity included both pre- and postcentral gyrus (Table 6).

Precentral connectivity was more extensive in the right hemisphere during repetitive tapping (Fig 7), whereas postcentral connectivity was more extensive in the left hemisphere during sequential learning (Fig 6).

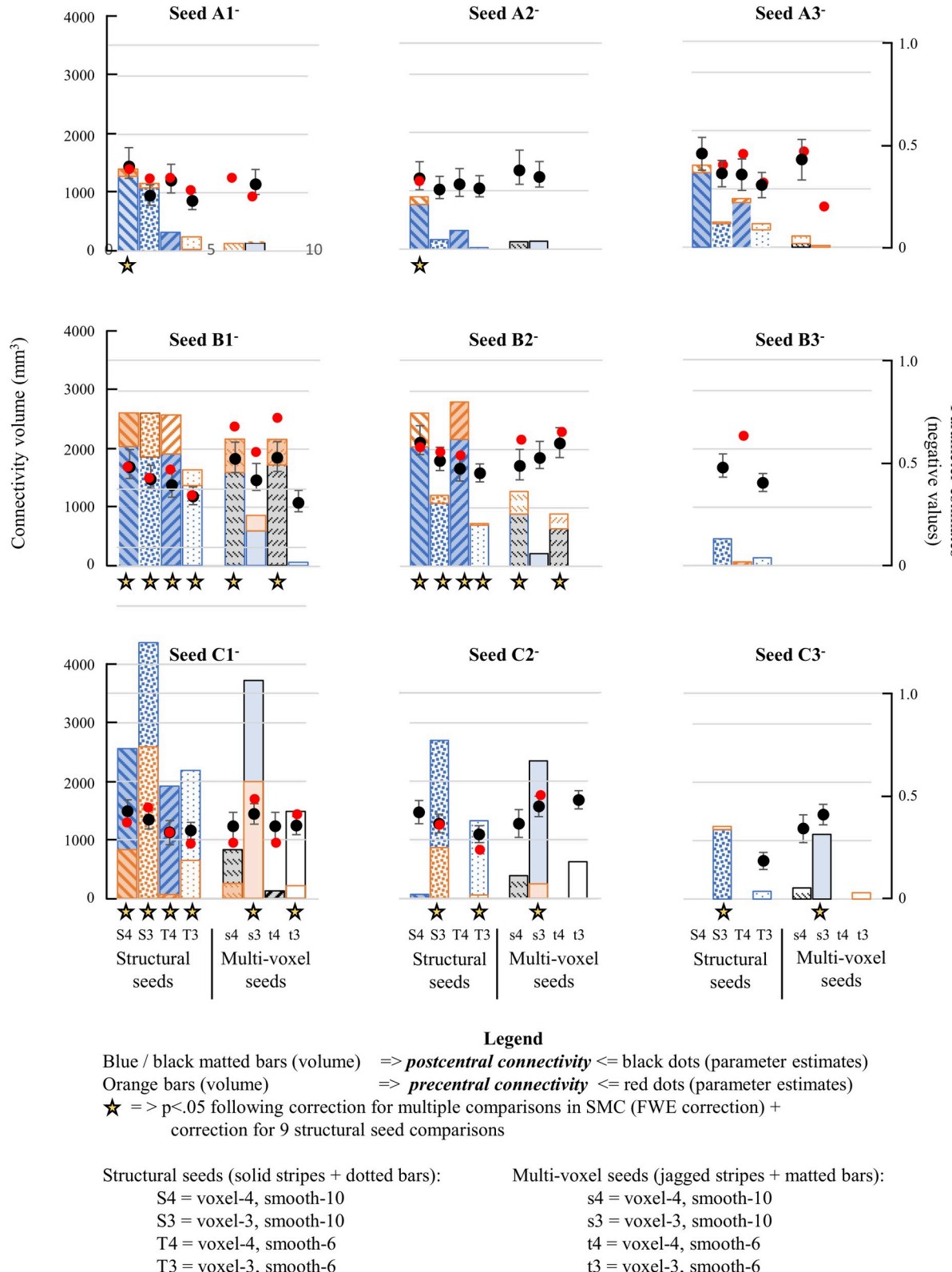

**Fig 5. Comparison of different methods of analysis on hippocampal connectivity during sequence learning.** Volume of connectivity and mean beta parameter estimates from eight methods of analysis at each region identified for structural seeds. Larger volumes of connectivity were elicited from structural seeds, particularly in medial (left) and posterior regions (bottom, see especially C1). Larger connectivity volumes

were generated from 3mm isovoxels in posterior regions but from 4mm isovoxels elsewhere; larger parameter estimates were also associated with the larger (10mm) smoother kernel among structural seeds. All connectivity volumes are shown from random effects analysis of the left hippocampus ($p<0.05$ after FWE correction for SMC hand representation). Postcentral and precentral volumes are stacked columns except at C1 and C2, where pre- and postcentral volumes were superimposed to maintain the same graph scaling.

**Table 5. Smoothing kernel effects on hippocampal connectivity from structural seeds during sequence learning.**

| 3mm isovoxels | | Smooth 6 | | | Smooth 10 | | | |
|---|---|---|---|---|---|---|---|---|
| *Structural* | coordinates | t-value | extent | volume | coordinates | t-value | extent | volume |
| A1⁻ (post) | (-39,-37,47) | 5.83 | 8 | 216 | (-36,-37,50) | 4.94 | 40[†] | 1080 |
| (pre) | (-30,-13,68) | 4.98 | 1 | 27 | (-30,-13,68) | 4.64 | 3 | 81 |
| A2⁻ (post) | (-36,-37,50) | 5.09 | 1 | 27 | (-33,-37,47) | 4.53 | 6 | 162 |
| (pre) | -- | -- | -- | -- | -- | -- | -- | -- |
| A3– (post) | (-33,-37,47) | 6.11 | 11[†] | 297 | (-33,-37,47) | 4.76 | 15 | 405 |
| (pre) | (-42,-22,44) | 5.59 | 4 | 108 | (-30,-13,68) | 4.20 | 1 | 27 |
| B1⁻ (post) | (-36,-37,50) | 7.38* | 51 | 1377 | (-36,-37,47) | 6.25* | 69 | 1863 |
| (pre) | (-30,-13,68) | 7.49* | 10 | 270 | (-33,-13,68) | 6.28* | 28 | 756 |
| B2⁻ (post) | (-39,-37,50) | 6.23 | 27[†] | 729 | (-36,-37,47) | 6.04* | 40 | 1080 |
| (pre) | -- | -- | -- | -- | (-33,-13,65) | 4.49 | 5 | 135 |
| B3⁻ (post) | (-39,-37,50) | 5.02 | 5 | 135 | (-36,-37,47) | 4.83 | 17 | 459 |
| (pre) | -- | -- | -- | -- | -- | -- | -- | -- |
| C1⁻ (post) | (-36,-34,47) | 8.07* | 81 | 2187 | (-33,-34,47) | 8.51* | 162 | 4374 |
| (pre) | (-30,-13,68) | 6.64* | 24 | 648 | (-33,-13,65) | 6.85* | 96 | 2592 |
| C2⁻ (post) | (-39,-31,47) | 7.95* | 49 | 1323 | (-33,-34,47) | 8.43* | 100 | 2700 |
| (pre) | (-33,-10,59) | 5.16 | 2 | 54 | (-36,-10,65) | 5.61 | 32[†] | 864 |
| C3⁻ (post) | (-36,-34,47) | 5.47 | 5 | 135 | (-33,-34,47) | 6.83* | 44 | 1188 |
| (pre) | -- | -- | -- | -- | (-36,-10,62) | 4.32 | 2 | 54 |
| **4mm isovoxels** | | **Smooth 6** | | | **Smooth 10** | | | |
| *Structural* | coordinates | t-value | extent | volume | coordinates | t-value | extent | volume |
| A1⁻ (post) | (-46,-28,58) | 4.16 | 4 | 256 | (-46,-28,58) | 3.86 | 20[†] | 1280 |
| (pre) | (-38,-12,62) | 3.73 | 1 | 64 | (-38,-12,62) | 3.65 | 2 | 128 |
| A2⁻ (post) | (-38,-36,50) | 4.11 | 5 | 320 | (-34,-36,50) | 3.71 | 12[†] | 768 |
| (pre) | -- | -- | -- | -- | (-34,-12,66) | 3.50 | 2 | 128 |
| A3⁻ (post) | (-34,-36,50) | 4.01 | 12 | 768 | (-34,-36,50) | 3.89 | 20 | 1280 |
| (pre) | (-34,-12,66) | 3.70 | 1 | 64 | (-34,-12,66) | 3.75 | 2 | 128 |
| B1⁻ (post) | (-38,-36,50) | 5.38* | 30 | 1920 | (-38,-32,46) | 4.42* | 32 | 2048 |
| (pre) | (-34,-12,66) | 6.20* | 12 | 768 | (-34,-12,66) | 5.38* | 9 | 576 |
| B2⁻ (post) | (-38,-36,50) | 5.60* | 34 | 2176 | (-38,-36,50) | 4.31* | 32 | 2048 |
| (pre) | (-34,-12,66) | 5.24* | 10 | 640 | (-34,-12,66) | 4.96* | 9 | 576 |
| B3⁻ (post) | -- | -- | -- | -- | -- | -- | -- | -- |
| (pre) | (-38,-12,62) | 3.68 | 1 | 64 | -- | -- | -- | -- |
| C1⁻ (post) | (-38,-32,54) | 4.66* | 30 | 1920 | (-38,-32,46) | 4.73* | 40 | 2560 |
| (pre) | (-38,-16,62) | 3.47 | 1 | 64 | (-34,-12,66) | 4.80* | 13 | 832 |
| C2⁻ (post) | -- | -- | -- | -- | (-38,-32,46) | 3.22 | 1 | 64 |
| (pre) | -- | -- | -- | -- | -- | -- | -- | -- |
| C3⁻ (post) | -- | -- | -- | -- | -- | -- | -- | -- |
| (pre) | -- | -- | -- | -- | -- | -- | -- | -- |

* Intensity threshold $p < .05$ after an additional FWE correction for evaluating 9 seed locations.

† Extent threshold $p < .001$ after an additional FWE correction.

**Table 6. Statistical effects of different methods of connectivity analysis across all anatomical locations during sequence learning.**

| | | Postcentral | | | Precentral | | |
|---|---|---|---|---|---|---|---|
| | | Volume difference[1] | Parameter difference[2] | p-value | Volume difference | Parameter difference | p-value |
| **Structural: Effect of voxel size** | S4 vs. S3 | 566.8 | -.077±.036 | .035* | 102.3 | -.022±.045 | .626 |
| | (smooth10) | **-2225.0** | | | **-1760.0** | | |
| | T4 vs. T3 | 614.8 | .011±.019 | .027* | 102.3 | -.010±.038 | .011* |
| | (smooth6) | **-575.0** | | | **-584.0** | | |
| **Effect of smoothing kernel** | S4 vs. T4 | 374.5 | -.134±.033 | .015* | 160.8 | -.028±.053 | .602 |
| | (voxel4) | | | | | | |
| | S3 vs. T3 | 789.0 | -.050±.011 | < .001* | 604.8 | -.106±.023 | < .001* |
| | (voxel3) | | | | | | |
| **Multi-voxel: Effect of voxel size** | s4 vs. s3 | 321.6 | -.014±.026 | .602 | 306.0 | .045±.049 | .379 |
| | (smooth10) | **-464.4** | | | **-1742.0** | | |
| | t4 vs. t3 | -54.0 | -.047±.043 | .299 | -- | -.161±.111 | .173 |
| | (smooth6) | **-1053.0** | | | **-216.0** | | |
| **Effect of smoothing kernel** | s4 vs. t4 | 64.0 | .049±.035 | .162 | 130.0 | .040±.057 | .494 |
| | (voxel4) | | | | | | |
| | s3 vs. t3 | 1503 | .003±.036 | .903 | 1026.0 | .004±.049 | .478 |
| | (voxel3) | | | | | | |
| **Structural vs. Multi-voxel** | S4 vs. s4 | 877.7 | -.05±.036 | .157 | 256 | -.045±.067 | .506 |
| | (vox4/sm10) | | | | | | |
| | S3 vs. s3 | 573.8 | .011±.018 | .153 | 438.8 | .053±.028 | .068 |
| | (vox3/sm10) | | | | | | |
| | T4 vs. t4 | 1603.3 | .134±.029 | < .001* | 457.3 | .125±.069 | .073 |
| | (vox4/sm6) | | | | | | |
| | T3 vs. t3 | 909.0 | .095±.030 | .003* | 351.0 | .178±.051 | .002* |
| | (vox3/sm6) | | | | | | |

[1]Mean differences in volume during group analysis across anatomical seed locations, specific to those with demonstrable connectivity across both conditions. The mean from posterior seeds is shown in boldface when comparing different voxel sizes to differentiate effects in posterior hippocampus.

[2]Mean differences in parameter estimates across anatomical seed locations. Negative values reflect larger inverse connectivity in the first condition.

Overall, the 10mm smoothing kernel provided better sensitivity for structural seeds to demonstrate both hippocampal connectivity and topography.

### Hippocampal connectivity: Functional seeds

Functional seeds showed unilateral connectivity during group analysis. Tapseed⁻generated right-hemispheric connectivity during the repetitive tapping task (Fig 8, red), whereas memseed⁻generated connectivity in the left hemisphere (yellow). Smoothing kernels of 6mm and 10mm showed no differences in connectivity during repetitive tapping or in postcentral gyrus during sequence learning, whereas the extent of precentral connectivity during sequence learning was larger with the 10mm kernel for both memseed⁻ (2880 vs. 1600 mm³) and memseed⁺ (1728 vs. 320 mm³).

## Discussion

This study evaluated effects of seed selection (structural vs. multivoxel vs. functional), voxel size (3mm vs. 4mm isovoxels), and smoothing kernels (0-10mm) on hippocampal-SMC

**Table 7. Smoothing kernel effects on hippocampal connectivity with SMC hand representation during repetitive tapping.**

| 4mm isovoxels (Bilat) | | Smooth 6 | | | Smooth 10 | | | |
|---|---|---|---|---|---|---|---|---|
| *Structural* | coordinates | t-value | extent | volume | coordinates | t-value | extent | volume |
| A1⁻ | (38,-24,46) | 4.23 | 4 | 256 | (34,-24,46) | 4.05 | 30[†] | 1920 |
| A2⁻ | (38,-24,46) | 5.44* | 47 | 3008 | (38,-16,54) | 4.58* | 54 | 3456 |
| A3⁻ | (38,-24,46) | 5.18* | 46 | 2944 | (30,-16,62) | 4.88* | 54 | 3456 |
| B1⁻ | (38,-24,46) | 4.56 | 15 | 960 | (38,-24,46) | 4.51* | 46 | 2944 |
| B2⁻ | (38,-24,46) | 4.51 | 14 | 896 | (38,-24,46) | 4.42* | 43 | 2752 |
| B3⁻ | (38,-24,46) | 4.59 | 18[†] | 1152 | (38,-24,46) | 4.17 | 43[†] | 2752 |
| C1⁻ | (38,-24,46) | 4.94* | 21 | 1344 | (34,-24,46) | 4.81* | 52 | 3328 |
| C2⁻ | (38,-24,46) | 3.56 | 2 | 128 | (34,-24,46) | 3.89 | 24 | 1536 |
| C3⁻ | (38,-24,46) | 3.59 | 1 | 64 | (34,-24,46) | 3.78 | 27 | 1728 |
| *Functional* tapseed⁻ | (42,-24,58) | 7.23 | 54 | 3456 | (30,-16,62) | 5.96 | 54 | 3456 |

* Intensity threshold $p < .05$ after an additional FWE correction for evaluating 9 seed locations.

† Extent threshold $p < .001$ after an additional FWE correction.

connectivity during group analysis; effects of voxel size (3-5mm) and smoothing kernels (0-14mm) on motor activation and connectivity during individual analysis were also evaluated. Voxels 4mm in size modestly improved sensitivity for detecting activation in SMC while maintaining selectivity for the hand representation; they also improved detection of connectivity from most regions of the hippocampus (except the posterior third of the hippocampus as compared to 3mm isovoxels). Smoothing kernels up to 10mm improved sensitivity for detecting hippocampal connectivity, allowing the topography of hippocampal connectivity to be mapped. Structural seeds showed greater sensitivity for connectivity than multivoxel seeds, but were more affected by the size of the smoothing kernel than functional seeds; the latter addressed individual variability by selecting voxels with maximal SMC connectivity within a hippocampal region. In contrast to hippocampal activation analysis, individual results suggest that larger (4mm) isovoxels and smoothing kernels improve connectivity by identifying the *temporal* pattern of activity as a composite from neighboring voxels.

### Effects of voxel size and smoothing kernel on cortical activation

For both repetitive tapping and sequence learning, voxel size had modest effects on SMC activation, with 4mm isovoxels showing a smaller statistical threshold and a larger activation volume. These effects were small, detected through direct within-subject comparisons, whereas group comparisons of the mean showed no effect. Similarly, random-effects activation revealed only minor differences from voxel size in the area of activation along cluster borders, with no significant differences in contrast magnitude (regardless of smoothing kernel).

The effect of smoothing kernel on cortical activation depended on the ROI. With the SMC mask, increases in the smoothing kernel from 6mm to 10mm resulted in the detection of much greater activation volumes; within the hand representation, this increase in activation volume was much less. This suggests the larger smoothing kernel more effectively reduced random variability (noise) from inactive regions within the SMC mask, reducing the mean difference in signal required to detect activation. This interpretation is consistent with observed decreases in the maxima t-value, contrast magnitude, and threshold.

During both individual and group analysis, the largest (10mm) smoothing kernel applied to 4mm isovoxels was most sensitive for detecting activation. This was not simply an overestimation of activation size from a large smoothing kernel, as suggested by others [15]; global

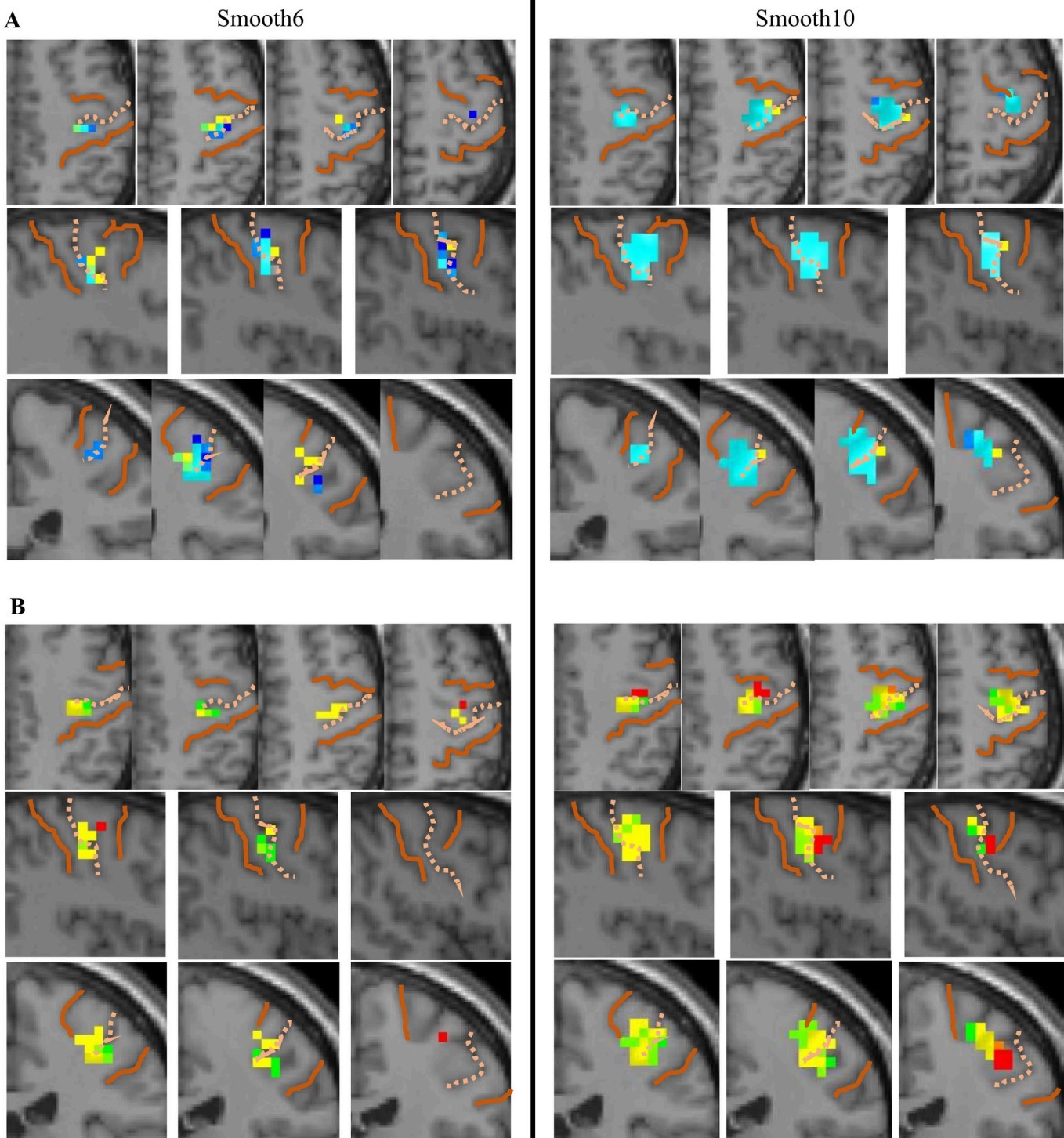

**Fig 6. Effects of smoothing kernels on hippocampal connectivity from structural seeds during repetitive tapping.** (A) Connectivity plotted from structural seeds along the medial/lateral axis of the hippocampus near its center, specifically, B1 (yellow, central-medial), B2 (cyan, central-middle), and B3 (dark blue, central-lateral). A larger area of connectivity was apparent with 10mm smoothing; a topographical organization was suggested with both smoothing kernels, although partly obscured with 10mm smoothing due to the enlarged area of connectivity from B2 (cyan). (B) Connectivity plotted from medial structural seeds along the longitudinal axis of the hippocampus, specifically, seeds A1 (red, anterior-medial), B1 (yellow, central-medial), and C1 (green, posterior-medial). The 10mm smoothing kernel generated a larger area of connectivity and a distinct topographical organization within each plane. Images illustrate random effects analysis of bilateral connectivity from 4mm isovoxels; SMC boundaries are demarcated with solid orange lines, with dotted lines demarcating the central sulcus.

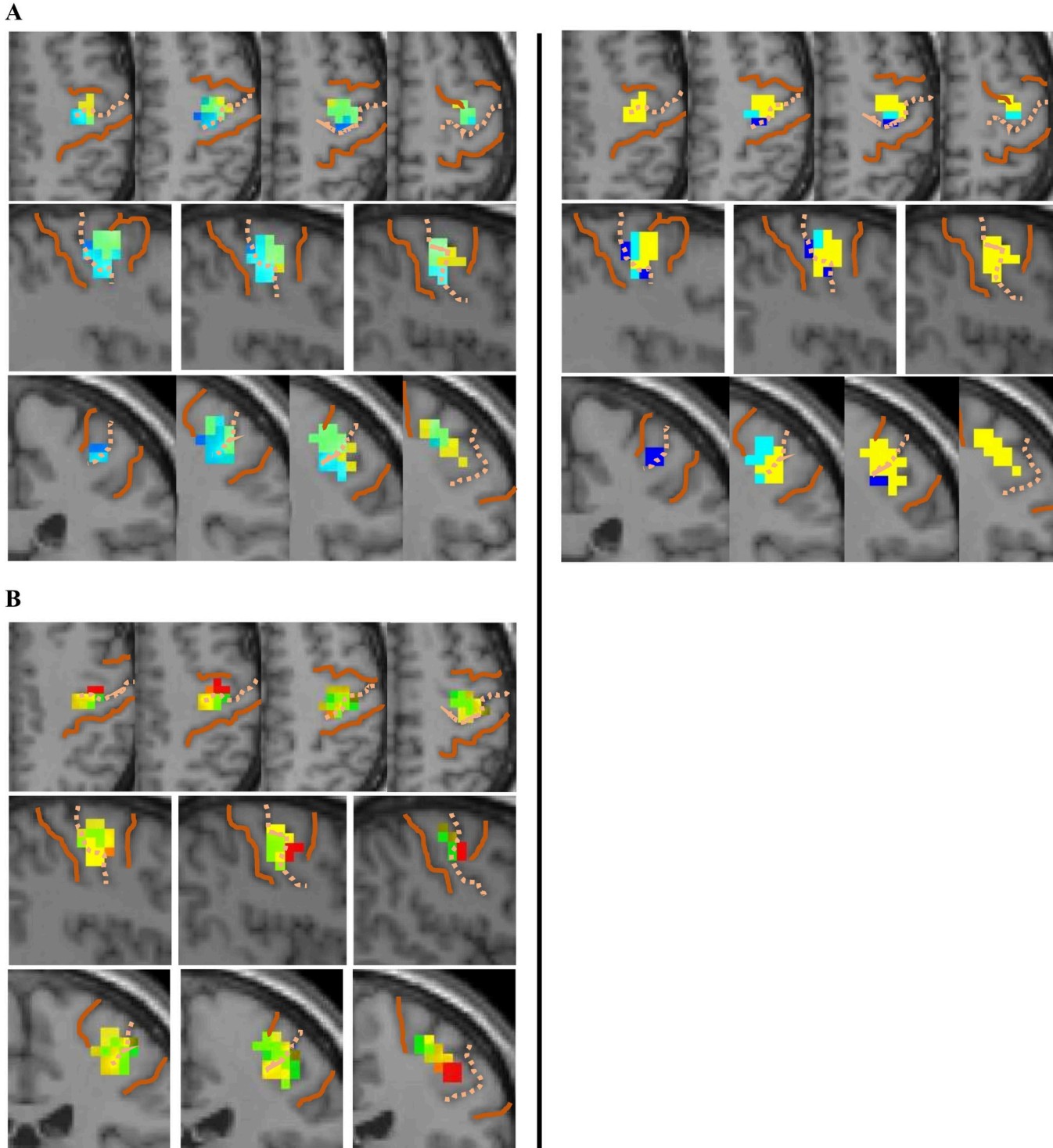

**Fig 7. Topographical hippocampal connectivity from individual voxels during repetitive tapping.** (A) Connectivity maps from three central voxels along the medial-lateral plane. Overlapping connectivity from adjacent voxel maps is shown on the left; differential connectivity is shown on the right (lateral > middle seed connectivity in dark blue, middle > medial seed connectivity in cyan, medial seed connectivity in yellow). Selected seeds are second, fourth, and sixth voxel from the medial edge of the hippocampus (from a breadth of six 4mm-isovoxels), representing structural seeds B1, B2 and B3. (B) Connectivity maps from three medial voxels along the longitudinal plane; anterior seed is red, middle is yellow, and posterior is green. Selected voxels are the second, fifth, and eighth from the anterior edge of the hippocampus (from a length of nine 4mm isovoxels), representing structural seeds A1, B1, and C1. Conventions as in Fig 4.

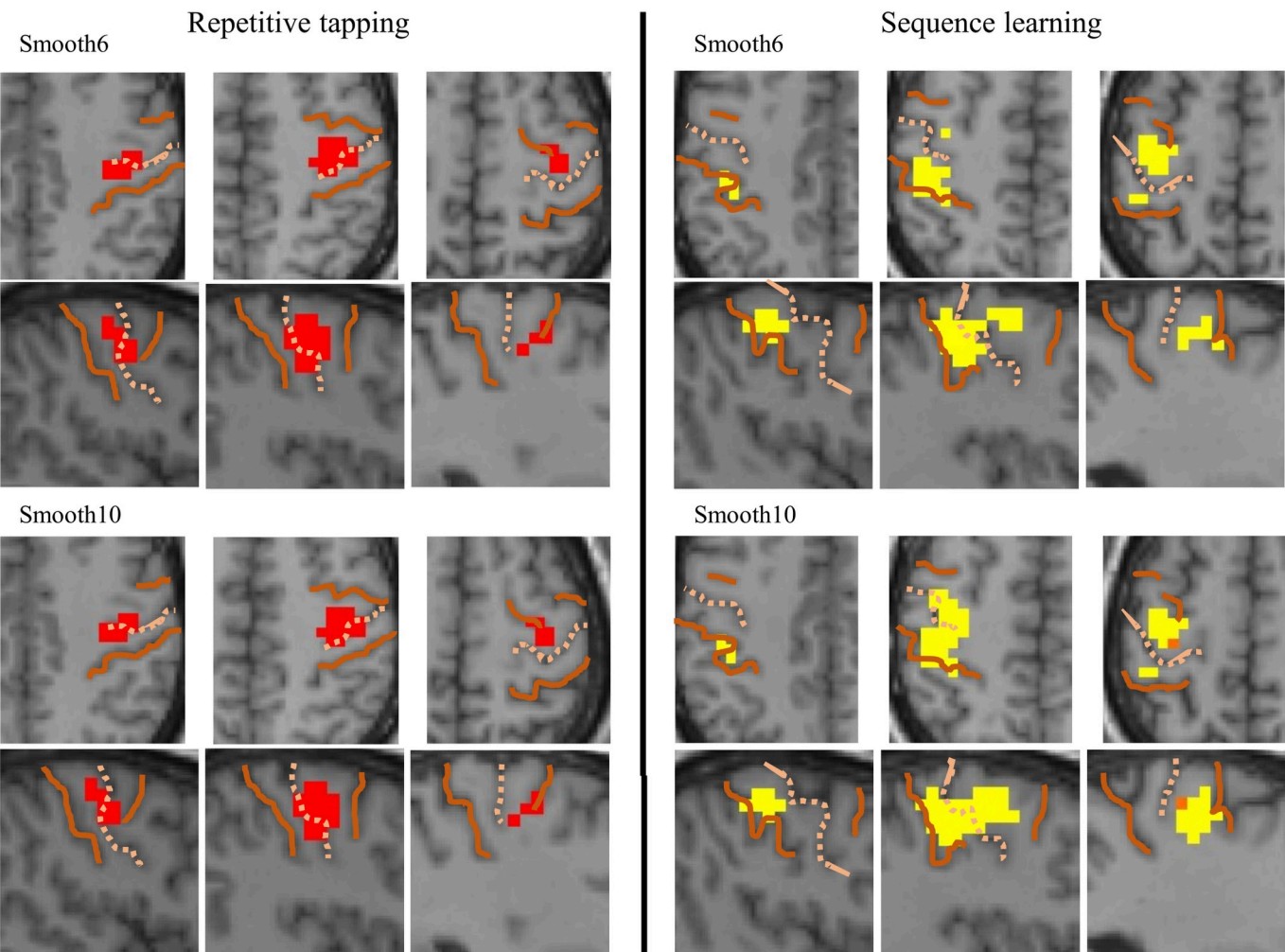

**Fig 8. Hippocampal connectivity from functional seeds showing motor activity.** Connectivity from functional seeds during repetitive tapping (red) or sequence learning (yellow) were similar in location and extent for both smoothing kernels.

analysis across both tasks demonstrated an even larger, bilateral area of activation consistent with task behavior [39]. With these processing parameters, the contrast estimate at the maxima from group analysis approached the mean value from individual analyses, suggesting this combination of voxel size and smoothing kernel provided the most accurate group representation of sensorimotor activation.

In analysis from a single subject, voxel-wise analysis of SMC suggested 4mm isovoxels with 10mm smoothing were optimal parameters for identifying the full extent of SMC activation. This smoothing kernel corresponded to the width of functional activation, even before smoothing.

## Effect of voxel size and smoothing kernel on hippocampal activation

During voxel-wise analysis in an individual subject, the largest effects of activation and de-activation were observed without smoothing in the smallest voxel size (3mm). With 4mm isovoxels, activation was slightly reduced in amplitude before smoothing; with larger voxel sizes or smoothing, no activation was observed. Similar to SMC activation, the width of functional

activation before smoothing was around 8-12mm. From this analysis, optimal parameters for hippocampal activation analysis appeared to require small voxels with little or no smoothing.

No significant hippocampal activation was observed during group analysis, regardless of voxel size or smoothing kernel.

### Effect of voxel size and analytical approach on hippocampal connectivity

Because direct point-to-point comparisons is not possible between different voxel sizes, effects were identified by comparing the volume of significant connectivity and maximal parameter estimates across different methods and smoothing kernels. Using a 3x3 matrix, all hippocampal regions were sampled, thereby allowing detection of regional effects.

Traditionally, either the entire hippocampus or a multivoxel subregion is specified as a seed for connectivity analysis [46–49]. Multivoxel seeds average hippocampal activity before looking for correlations elsewhere in the brain; this differs from our structural seed alternative, which identified the magnitude of correlations from each voxel within a hippocampal region, then averaged these connectivity maps. The method of analysis showed inconsistent effects on the maximal parameter estimate from a region; however, greater connectivity volumes were consistently observed from structural seeds in all regions, indicating this approach improves sensitivity for detecting connectivity. Indeed, multivoxel seeds failed to demonstrate significant connectivity in anterior hippocampus. Furthermore, the topographic organization of hippocampal connectivity only emerged with structural seed analysis.

In posterior hippocampus, 3mm voxels generated greater connectivity volumes than 4mm voxels, particularly in posteromedial hippocampus. This likely reflects both anatomical and functional properties of the hippocampus. The posterior hippocampus is thinner (four 4mm-voxels wide, compared with six in middle and anterior regions); furthermore, posteromedial hippocampus may be functionally more uniform. More lateral regions include positive connectivity, particularly in middle-to-anterior regions (e.g., see S2 Fig from [39]). A sharp transition between functionally distinct regions may thus be requisite for smaller voxels to demonstrate greater connectivity. By contrast, 4mm isovoxels generated greater connectivity volumes in anterior and central regions of the hippocampus, as well as generating larger parameter estimates at structural seed maxima (Fig 5 and Table 6).

In summary, structural seeds were more effective for demonstrating hippocampal connectivity. Except in posterior areas, perhaps due to a transition to a functionally distinct area, 4mm generated more extensive connectivity maps than 3mm voxels. In this study, a topographical organization to hippocampal connectivity was only evident with the structural seed analysis using 4mm voxels.

### Effect of smoothing kernel on hippocampal connectivity

This study showed different patterns of inverse hippocampal connectivity with SMC during sequence learning and repetitive tapping: the left hippocampus generated both pre- and postcentral connectivity maxima in left SMC during sequence learning, whereas bilateral hippocampal seeds generated a single connectivity maximum in the right pericentral gyrus during repetitive tapping. The size of the smoothing kernel showed similar effects on connectivity during both tasks.

Structural seeds demonstrated larger clusters of SMC connectivity with the larger kernel in both motor tasks. A connectivity map from a structural seed represents the mean connectivity from every voxel within the designated region of hippocampus, so larger connectivity clusters reflect the effectiveness of the smoothing kernel in reflecting local regional activity (see also [22]).

With the 6mm smoothing kernel, a topographic pattern from structural seeds was evident along the medial-lateral axis of the hippocampus, but not the longitudinal axis, where small and disparate clusters were generated with no apparent topography. With the 10mm kernel, a topographic pattern was partially obscured along the medial-lateral axis, due to extensive overlap in connectivity from adjacent seeds, yet a topography along this axis was still evident from connectivity of individual voxels. Furthermore, the 10mm smoothing kernel was needed to demonstrate topography along the longitudinal axis of the hippocampus.

The size of the smoothing kernel may be a factor in detecting functional heterogeneity within the hippocampus. A resting state connectivity study using a small (3mm) smoothing kernel failed to find functional heterogeneity along the anterior-posterior axis [50], whereas a study of connectivity from the hippocampus to cortical regions during episodic memory found functional heterogeneity along the same axis using a larger (5mm) smoothing kernel [28].

In summary, the 10mm smoothing kernel was advantageous for demonstrating hippocampal topography in its connectivity with SMC.

## Topography of hippocampal connectivity

Similar to previous studies, hippocampal connectivity in this study showed a topographical organization along both the long anterior-posterior axis [51–54] and medial-lateral axis [51].

In the current study, hippocampal connectivity with SMC was preferentially seen within the SMC hand representation for both motor tasks, with location and laterality similar to group activation by the tasks. Group activation and inverse connectivity were both limited to the left SMC during sequence learning (reflecting right-handed movements) and the right SMC during repetitive tapping (despite bilateral hand movements). Furthermore, activation and connectivity both covered the breadth of the postcentral gyrus within the hand representation during sequence learning, whereas activation and connectivity were both centered in precentral gyrus during repetitive tapping, with postcentral connectivity limited to the region adjacent to the central sulcus. Whether these task differences arose from differences in behavioral requirements, hand dominance (right-handed subjects), or hemispheric differences in motor function is unknown.

The role of hippocampal topography to SMC function is unclear. Within the hand area of the precentral gyrus, overlapping but distinguishable regions are involved in wrist and individual finger movements [35, 55–57]; multiple representations for each digit may differentially reflect flexion and extension movements [38] or different degrees of movement complexity [58]. Similarly, somatotopy for individual fingers in postcentral gyrus are overlapping but distinguishable, with multiple representations across the anterior-posterior width of the gyrus [59–61]. Methods here did not allow mapping onto individual finger representations, or even flexion vs. extension, so the functional significance of the hippocampal topography observed here is unclear.

Different cognitive functions have been suggested for anterior vs. posterior hippocampus, differences such as encoding vs. recall [62, 63], internal vs. external attention [64], gist/conceptual vs. detailed/spatial information [65–67], and pattern completion vs. separation [68]. In the current study, the topographical pattern of connectivity observed from anterior vs. posterior hippocampus is inconsistent with these roles, particularly as the repetitive tapping task required paced, volitional movements without learning or memory recall [39]. Topographical connectivity with different regions within the pre- or postcentral hand representation suggest a role more directly involved in motor control, perhaps coordinating cognitive and motor functions to facilitate and suppress muscle actions required for the correct timing of finger movements [40, 57].

## Validity and limitations

This study used an effective connectivity technique, which examines the directional influence of on area on another. PPI is directional but also task-specific [44], in this study identifying hippocampal influences on SMC activity during a specific motor condition. Hippocampal influences were identified during sequence learning (which involved memory), but also during repetitive tapping (which did not). These influences differed in their hippocampal origins (primarily left hippocampus for sequence learning, bilateral for repetitive tapping), but also their cortical targets (left SMC for inverse connectivity during sequence learning, right SMC for repetitive tapping). With such specificity, the question arises whether the same methods for optimizing results in this study apply elsewhere.

As noted in the introduction, most hippocampal studies use small voxel sizes (1-3mm) and smoothing kernels (0-6mm). In the individual analysis, these parameters were effective for localizing hippocampal activation. With 3mm isovoxels and no smoothing, maximal activation in a representative individual was observed medially with deactivation laterally; deactivation was not observed with 4mm voxels, and neither activation nor deactivation were observed with 5mm voxels. Because hippocampal activation was not noted during group analysis for either 3mm or 4mm voxels, however, the practical relevance of this for group studies is unclear. By contrast, hippocampal-SMC connectivity was better observed with 4mm isovoxels and 10mm smoothing for functional seeds in this individual, mirroring the trend observed in group analysis.

This paradox can be understood from the individual analyses presented here. The temporal eigenvariate waveform used in PPI connectivity analysis depended on both the voxel size and smoothing kernel; smoothing reduced activity from the central voxel while increasing the contribution from neighboring voxels. Connectivity was detected when SMC activity matched an interactive term derived from this waveform. In the current study, this was optimal with 4mm isovoxels and a 10mm smoothing kernel, suggesting a multivoxel functional unit within the hippocampus. Without smoothing, connectivity maxima in SMC were weak and non-significant.

In comparing effects of different methods, this study largely focused on volume and the magnitude of beta estimates (see especially Fig 5). These results were suggested to reflect the sensitivity of different methods. Connectivity matches the temporal pattern of SMC activity to a single waveform, as defined by the task-specific pattern of activity in the hippocampal seed. Unlike activation analysis, increasing the size of voxels and the smoothing kernel did not indiscriminately increase the area of SMC that matches this criterion; changes in parameters instead changed the waveform of seed activity. The volume of connectivity detected, then, depended only on the amplitude and variance of SMC activity that appropriately matched the seed's waveform, given the parameters used for analysis.

Reports of connectivity or activation depended on the intensity and volume of effects–and sometimes both to include corrections for multiple comparisons. In Fig 5, connectivity volumes and beta estimates from diverse methods were displayed based on statistical criteria applied before correction for multiple comparisons. Findings with some methods would not meet such a correction, so would not normally be reported; these methods lacked the *sensitivity* to detect connectivity from hippocampal regions that other methods do not. Arguably, the volume of connectivity detected can be used as a measure of sensitivity, considering the nature of the criteria that must be met (as detailed above), since more voxels showed an effect that otherwise would not be detected [1].

Parameters shown to be optimal for this study should not be considered universally applicable to all hippocampal studies, as they necessarily depend on the nature of the hypothesis;

parameters optimal for connectivity analysis in this study, for example, were not the same as those identified for analysis of hippocampal activation in an individual. Importantly, the current study shows that identifying hippocampal connectivity effects on cortical areas require adequate parameters for characterizing cortical function as well as that of the hippocampus.

Three points about optimal parameters may be emphasized. First, traditions that limit the size of voxels and smoothing kernels are not necessarily optimal. Second, parameter affects connectivity analysis different than activation analysis, despite similar conventions for reporting results. Third, statistical results using non-optimal parameters show real effects, but may understate their extent. Hippocampal connectivity studies acquired with different resolutions and involving other cortical areas are needed to elucidate how widespread the optimal parameters from this study may be applied.

## Practical considerations for future studies

Many hippocampal studies have used small voxels and smoothing kernels, believing these processing parameters more accurately reflect true hippocampal activity. The current study suggests this may not always be the case, particularly when examining hippocampal connectivity with cortical areas. This study specifically examined the effects of voxel size and smoothing kernel on hippocampal activation and connectivity with SMC; however, results may also apply to other hippocampal connectivity studies, as observed effects on cortical activity from changes in processing parameters are consistent with previous studies (as noted earlier).

Results also indicate greater sensitivity can be obtained with structural seeds, rather than conventional multivoxel seeds. For structural seed analysis, connectivity is calculated from each voxel of the hippocampus, then a mean connectivity map is calculated from all voxels in a region. This approach can map connectivity from the entire hippocampus, particularly advantageous when the precise location of a functional region is unknown.

Paired t-tests comparing cortical activation with 4mm vs. 3mm isovoxels showed lower thresholds and larger volumes of activation from larger voxels, with the additional volume expanding the edges of an activation cluster. Similarly, larger smoothing kernels decreased the threshold and increased the volume of cortical activation. Even small differences in activation between 8mm and 10mm smoothing kernels were significant during sequence learning, although these differences did not reach significance for 4mm voxels during repetitive tapping.

Parameters optimal for detecting SMC cortical activation also improved hippocampal connectivity (10mm smoothing kernels and 4mm voxels, except from the posterior third of the hippocampus). This similarity suggests better detection of cortical activity might allow better correlation with hippocampal activity. Voxel size and smoothing also affected the temporal activity pattern of the hippocampus seed, however, which should remind us that the effectiveness of processing parameters in a connectivity study necessarily reflects both cortical and local hippocampal properties.

Because the size and structure of the hippocampus is affected by a wide array of factors, including age, exercise, depression, and stress [69–74], its functional localization must be variable. Functional seeds sought to reduce effects of individual variability during group analysis. Restricted to structural regions showing significant connectivity during group analysis, the voxel showing maximal connectivity within the entire SMC was identified. Using this ROI during seed selection, the selection of functional seeds did not bias the outcome; connectivity maxima used for seed selection were scattered, located in both hemispheres and often outside the hand representation [39]. These functional seeds nonetheless showed selective connectivity within the hand region during group analysis. The current study showed that the sensitivity of functional seeds was minimally affected by the size of the smoothing kernel, suggesting

smoothing mitigates the effects of individual variability during group analysis, but also that sensitivity for demonstrating connectivity with non-optimal parameters can be improved by identifying functional seeds.

The loci of functional seeds might be suggested from hippocampal maxima in those studies where activation can be reliably identified. Information carried in the spatiotemporal pattern of hippocampal activity does not always reflect the changes in overall activity measured by activation analysis [75–77], however, and connectivity analysis compares moment-by-moment activity in two regions. Because activation within the hippocampus reflects functional activity, its presence can guide seed placement in connectivity studies, yet non-activated regions may also generate connectivity through their temporal properties. As such, a method that is not limited to *a priori* seed selection based on activation or local signal-to-noise can be appropriate for studies of hippocampal connectivity.

This article may be considered a methodological case example, demonstrating that processing parameters commonly used to study activation within the hippocampus may not be optimal for identifying its influence on cortical areas. Regardless of methodology, however, a meaningful description of the variability and effect of confounds on the resulting connectivity maps and inferences should always be considered; connectivity measurements reflect the net influence of all factors affecting hippocampal activity and its connections.

## Conclusions

This study shows that processing fMRI data with a larger voxel size (4mm) and smoothing kernel (8-10mm) can, at least in some cases, improve sensitivity to hippocampal activity and its connectivity with cortical areas; optimizing these processing parameters in this study uncovered topography in hippocampal connectivity along two axes. Structural seeds that represent the mean connectivity from all voxels in a hippocampal region may provide better sensitivity than traditional multivoxel seeds, and have the additional advantage of mapping connectivity throughout the hippocampus.

## Supporting information

**S1 Fig. Overview of data processing and analysis.** Data was resampled with 3mm, 4mm, and 5mm isovoxels following normalization. For each voxel size, data was smoothed and analyzed separately across a range of smoothing kernels.
(TIF)

**S2 Fig. Volume of activation in both motor tasks as a function of smoothing kernel and voxel size.** Within limits, larger smoothing kernels generated larger volumes of activation during group analysis regardless of voxel size. Maximal volume was typically generated with smoothing kernels of 8-12mm.
(TIF)

**S3 Fig. Spatial distribution of hippocampal activation from individual analysis.** (A) Effects of smoothing on activation magnitude from adjacent 3mm isovoxels; no smoothing (blue) and smoothing kernels of 6mm (orange) and 10mm (gray) are shown. Depending on similarity of responses in neighboring voxels, smoothing had minimal effects on peak activation or deactivation (black arrows) or greatly affect response amplitude (green arrows); the red arrow shows peak activation for the entire hippocampus. Summarized across all individuals, activation was elevated 9-12mm surrounding the peak, maintained with smoothing during repetitive tapping but eliminated during sequence learning. (B) Effects of smoothing on activation magnitude from adjacent 4mm isovoxels. Effects of smoothing again depended on similarity of responses

in neighboring voxels; before smoothing, activation was elevated 12-16mm surrounding the peak, reduced then eliminated with larger smoothing kernels for both repetitive tapping and sequence learning.
(TIF)

## Acknowledgments

The author wishes to thank the Center for Advanced Imaging (CAI) at NorthShore University HealthSystem for its administrative support.

## Author Contributions

**Conceptualization:** Douglas D. Burman.

**Data curation:** Douglas D. Burman.

**Formal analysis:** Douglas D. Burman.

**Investigation:** Douglas D. Burman.

**Methodology:** Douglas D. Burman.

**Project administration:** Douglas D. Burman.

**Visualization:** Douglas D. Burman.

**Writing – original draft:** Douglas D. Burman.

**Writing – review & editing:** Douglas D. Burman.

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
