## [Decision Letter · Decision Letter 0]

9 Apr 2021

PONE-D-20-38639

Topography of hippocampal connectivity with sensorimotor cortex revealed by optimizing smoothing kernel and voxel size

PLOS ONE

Dear Dr. Burman,

Thank you for submitting your manuscript to PLOS ONE. After careful consideration, we feel that it has merit but does not fully meet PLOS ONE’s publication criteria as it currently stands. Therefore, we invite you to submit a revised version of the manuscript that addresses the points raised during the review process.

The reviewers recognized the relevance of your study, but they also highlighted important points that should be addressed (see the detailed comments below). In particular there are two main problems from my point of view. First, to explore the effect of voxel size, it would be more accurate to compare distinct datasets, acquired with different raw voxel size (at least high-resolution vs low-resolution voxels), while keeping the other parameters equal (e.g. repetion and acquisition times, number of volumes). Considering the same dataset but preprocessed to generate different voxel sizes might bias the observed results. Second, the metric that was used to compare the results for the different voxel sizes and smoothing kernels might not be enough to provide a reliable conclusion.  

We look forward to receiving your revised manuscript.

Kind regards,

Jessica Dubois, Ph.D.

Academic Editor

PLOS ONE

Journal Requirements:

We note that you have stated that you will provide repository information for your data at acceptance. Should your manuscript be accepted for publication, we will hold it until you provide the relevant accession numbers or DOIs necessary to access your data. If you wish to make changes to your Data Availability statement, please describe these changes in your cover letter and we will update your Data Availability statement to reflect the information you provide.

Reviewers' comments:

Reviewer's Responses to Questions

**Comments to the Author**

1. Is the manuscript technically sound, and do the data support the conclusions?

Reviewer #1: Partly

Reviewer #2: Partly

2. Has the statistical analysis been performed appropriately and rigorously? 

Reviewer #1: No

Reviewer #2: No

3. Have the authors made all data underlying the findings in their manuscript fully available?

Reviewer #1: Yes

Reviewer #2: No

4. Is the manuscript presented in an intelligible fashion and written in standard English?

Reviewer #1: Yes

Reviewer #2: Yes

5. Review Comments to the Author

Reviewer #1: The author assesses voxel size and smoothing kernel choices to optimise both activation and connectivity of the hippocampus and the motor cortex. This estimation is relevant as the hippocampus is a small brain region with a higher sensitivity to noise and close to other regions of the temporo-mesial circuitry. I believe this work and similar investigation in our field is truly relevant while being difficult endeavour. Indeed, such evaluation need to be precise and use the full knowledge of possible artefacts, state of the art methodology and noise source to enable other researcher in the field to reuse this knowledge, generalize it and possibly have a pipeline to apply to their own dataset to estimate the optimal choice of parameter for their analysis. This investigation do not have a relevant metric that would enable to validate optimal choice of parameters.

In this regards I have a series of major comments that could help reframe the investigation to make the study relevant to the field.

The dataset used in this study is acquired at 3.4mm and either upsample to 3mm or downsample to 4mm. Unfortunately, these choices are not comparable as the upsampling preserve all the available information, while the downsample loose part of the information by applying a dry average. The interpretation is not related to bigger or smaller voxel anymore. I would suggest comparing 3mm with a small smoothing kernel to the 4mm data.

Additionally, as the data do not allow an investigation of what other studies are using which is lower size voxel (1-3mm as it is also mentioned by the author in the introduction), one way to assess some voxel size generalization would be to also propose a 5mm resampling as a theoretical comparison. This would enable to assess if the trend of increased voxel size effects replicate. However, having a second dataset with lower voxel size would strengthen the generalizability for hippocampal studies.

Regarding smoothing choice, if small regions are activated, large smoothing kernels may systematically bias or even obscure evidence of underlying activation (Friston et al., 1994). Spatial smoothing could systematically alter the localization of activation foci. This is what the investigation is showing: larger smoothing provides a larger “activation” area with smaller amplitude. See the figure 2 with the 6 to 10mm smoothing comparison. While a smaller smoothing enables to show a larger activation in a specific locus, a bigger smoothing would resemble a larger, more average and unspecific response. This does not mean we are learning more about the hippocampal response.

Related to this point of the meaning of the metric that is used in this estimation: While the task used might, as propose the author, be more robust and might display less individual variability, the link between the sensorimotor cortex and the hippocampus is not well described and can be challenging to use as a ground truth to evaluate parameters. On top of this, assuming that larger activation or connectivity (line 346) “reflect the sensitivity of the method” is not supported by the literature. Another metric related to the reproducibility of the results need to be used.

Additionally, to avoid this investigation to be dataset specific and in order to generalize to other studies, this estimation could be done in relation to data characteristic. Mainly I am thinking of signal-to-noise map of individual data. The hippocampus is localized in regions with higher noise component, but this is not described for this dataset and especially not put in relation to the optimization. The idea could be to assess the reproducibility of results using a voxel size/smoothing optimization in relation to level of local noise. How does noise affect the response and reproducibility of the activation/connectivity ? Knowing the signal to noise map of an acquisition: how small can we go in voxel and smoothing to reduce noise but keep an optimal local specificity?

Another methodological point for this study to be relevant in the field of hippocampal study is the choice of the segmentation of the hippocampus. As we see in the functional seed analysis, where we maximise the regional response strength: voxel size and smoothing do not affect the results much. Splitting the hippocampus into structural regions should also be optimized to reduce noise in the result by itself: did the author consider other, specifically individual-based segmentation and subfield segmentation? One example is the SACHA module (http://brainvisa.info).

Last methodological point is the interpretation of the connectivity multivoxel versus ROI average comparison. A direct comparison between the two approach of connectivity might not be relevant as this choice is dependent on the question that is under investigation. Using radius around a local maximum is used from previous study / literature generalization and to account for variability and noise around a reference locus. This should not be used to represent a ROI and so cannot be compare to an average or weighted average or other time series extraction from a ROI. Not the same amount of information is represented. This choice of atlas/segmentation/time series extraction are described in connectivity analysis guideline literature. This is an example https://pubmed.ncbi.nlm.nih.gov/23583357/

Minor comments :

Line 121 : two datasets are mentioned for 3 and 4mm, is there two independents dataset or is it the same dataset copied with two different sampling choice ? This was unclear from the text.

Line 134: the reference of WFUPickatlas toolbox is missing and proposed only later in the text.

Line 179 : “To minimize asymmetry effects, comparisons in connectivity across voxel sizes were limited to the left hippocampus.” This is confusing because left and right analysis are proposed in the results.

Fig 2 is missing brain slice numbers.

Fig 3 could use a bar stack code legend. The figure description talks about dash and dot and red/blue, but there is no explanation for grey, full colour or not, small or big dash, small or big or regular dots.

Reviewer #2: This study aimed to identify optimal processing parameters of smoothing the hippocampus during functional connectivity by manipulating already acquired data. The author showed that larger voxels and smoothing kernels improve the detection with regards to hippocampal activations and connectivity. I think The study has incremental findings to the field and the results were obvious to the researchers especially with regards to the usage of a larger smoothing kernel. This study, however, applied multiple tests and analyses that were not needed, in my opinion, and made the attention distracted.

The introduction is perfect. However, some studies are missing from being cited or not mentioned. For example, some studies investigated the effects of smoothing on resting-state functional connectivity, which is relevant to this study as the basic acquisition is the same. Also the study did not show any discussion about PPI...and why in particular it was used.

The abstract should identify PPI.

I also suggest the author go throughout the whole study and emphasis that the connectivity here was effective connectivity rather than just functional connectivity (to differentiate it from resting-state connectivity)

Also, the authors wrote "This study identified optimal processing parameters" while what I can see is they just focused on smoothing.

The methods were written acceptably. However, I had to read it several times to understand the details mentioned. I strongly suggest the authors to include a demographic summarising the methods including the performed task.

It seems to the reader at first that the task is a pure motor task but then it seems also it has memory components.

The voxel size of the acquisition is large. This is surprising giving the aim of focusing on the hippocampus. Also manipulating this voxel size in the preprocessing so that it goes to be isotropic or larger/smaller than what it seems make wondering about the accuracy and the meaning of this? For example, it could be much better and acceptable if the author acquired at least two sets of data with high resolution and low-resolution voxels. In this, the representations of the underlaying measured signals is not affected by the changes of the voxels at the preprocessing steps.

The kernels of smoothing were selected based on what? Also I see in the tables and figures that 0 smoothings was tested while nothing in the method speaks about this.

The statistical tests using excel are very questionable. The author could test these using SPM or any other functional software by dealing of this dataset as different conditions or groups.

"For 3mm isovoxels, 278 voxels were identified in the left hippocampus" How about the right side? and why not mentioned?

I think the discussion is acceptable. However methodological considerations of the study (not other studies) should be discussed

6. PLOS authors have the option to publish the peer review history of their article (what does this mean?). If published, this will include your full peer review and any attached files.

Reviewer #1: **Yes: **Roselyne J. Chauvin

Reviewer #2: No

---

## [Author Response · Author response to Decision Letter 0]

1 Jun 2021

To explore the effect of voxel size, it would be more accurate to compare distinct datasets, acquired with different raw voxel size (at least high-resolution vs low-resolution voxels), while keeping the other parameters equal (e.g. repetion and acquisition times, number of volumes). Considering the same dataset but preprocessed to generate different voxel sizes might bias the observed results. 

The same dataset was used to avoid introducing variability in subject performance and brain activity across task runs (issues of replication); by using the same dataset, even small differences must result from resampled voxel sizes. To minimize sampling bias, upsampling and downsampling occurred after preprocessing steps (i.e., after slice timing correction and realignment), and the data was resampled during the normalization stage. Acquisition parameters used in this study provided a good balance between temporal and spatial resolution while providing whole-brain coverage; comparison with other studies shows these parameters were not unusual (as detailed below).

The metric that was used to compare the results for the different voxel sizes and smoothing kernels might not be enough to provide a reliable conclusion. 

Volume and intensity differences are standard metrics for identifying group effects, both for activation and connectivity. The reliability of these group measures depends on their relationship to observed effects among individuals. The influence of voxel size and smoothing kernel on activation and its timecourse has been added from a representative individual, examined in sensorimotor cortex and the hippocampus. Findings suggest optimal parameters for activation and connectivity are not the same due to different effects of parameters on overall activation vs. the temporal pattern of hippocampal activity. Implications for determining the optimal combination during connectivity analysis are discussed in a new section, Validity and limitations.

If applicable, we recommend that you deposit your laboratory protocols in protocols.io to enhance the reproducibility of your results. Protocols.io assigns your protocol its own identifier (DOI) so that it can be cited independently in the future. 

A detailed protocol and SPM12 batch files have been deposited with NITRC, along with a link from the SPM12 extensions page; this is now specified in the methods. These batch files provide individual and structural seed PPI analysis for the entire hippocampus, using slice-time corrected, normalized, smoothed 4mm data from a lab’s activation analysis. 

We note that you have stated that you will provide repository information for your data at acceptance. Should your manuscript be accepted for publication, we will hold it until you provide the relevant accession numbers or DOIs necessary to access your data. If you wish to make changes to your Data Availability statement, please describe these changes in your cover letter and we will update your Data Availability statement to reflect the information you provide.

All data is available without restriction, and the dataset is available for download: Burman DD. Functional MRI dataset during passive visual stimulation and two motor conditions (sequence learning, paced repetitive tapping). NITRC 2019.

This is also noted in the methods and bibliography. 

Reviewer #1: The author assesses voxel size and smoothing kernel choices to optimise both activation and connectivity of the hippocampus and the motor cortex. This estimation is relevant as the hippocampus is a small brain region with a higher sensitivity to noise and close to other regions of the temporo-mesial circuitry. I believe this work and similar investigation in our field is truly relevant while being difficult endeavour. Indeed, such evaluation need to be precise and use the full knowledge of possible artefacts, state of the art methodology and noise source to enable other researcher in the field to reuse this knowledge, generalize it and possibly have a pipeline to apply to their own dataset to estimate the optimal choice of parameter for their analysis. This investigation do not have a relevant metric that would enable to validate optimal choice of parameters.

Voxelwise analyses to examine effects of voxel size and smoothing kernel on intensity and temporal eigenvariate waveforms have been added from individual subjects; the relationship of these results to the group analysis is discussed in the new section, Validity and limitations. One size does not fit all, but when examining hippocampal influences on cortical regions, parameters appropriate for cortical activation should be considered, as activity in the cortical functional unit will reflect its inputs. Understanding differences in the effects of smoothing on connectivity vs. activation is key to choosing optimal parameters. 

Statistics are provided for the volume and intensity of effects from group analysis, which are the metrics traditionally used to report activation and connectivity. Additional analysis suggests these measures accurately reflect the underlying processes seen in individuals.

In this regards I have a series of major comments that could help reframe the investigation to make the study relevant to the field.

The dataset used in this study is acquired at 3.4mm and either upsample to 3mm or downsample to 4mm. Unfortunately, these choices are not comparable as the upsampling preserve all the available information, while the downsample loose part of the information by applying a dry average. The interpretation is not related to bigger or smaller voxel anymore. I would suggest comparing 3mm with a small smoothing kernel to the 4mm data.

The claim that downsampling to 4mm but not upsampling to 3mm loses information is inaccurate. A functional dataset provides a spatio-temporal map of the echo planar signal in space; due to small movements of the brain, a small amount of brain tissue adjacent to any single brain voxel is sampled by the head coil for part of the time series. This additional craniospatial signal information justifies upsampling, but is also available along the edges of a voxel during downsampling. Furthermore, resampling was applied during the normalization process, where SPM expands the volume of the brain by 10% to minimize loss of signal. An in-plane voxel size of 3.4mm thus becomes 3.74; with the additional spatial information available, downsampling to 4mm affords little distortion. The comparable nature of the data is evident in a new figure that compares time courses of signal intensity with different voxel/smoothing combinations. With 10mm smoothing, the waveforms for 3mm and 4mm are virtually identical. 

Results for 3mm and 4mm voxels for activation, connectivity, and eigenvariate time courses of signal intensity can be compared from graphs across a range of smoothing kernels (see Figs 1-4 and Supplemental Figure 2). 

Additionally, as the data do not allow an investigation of what other studies are using which is lower size voxel (1-3mm as it is also mentioned by the author in the introduction), one way to assess some voxel size generalization would be to also propose a 5mm resampling as a theoretical comparison. This would enable to assess if the trend of increased voxel size effects replicate. However, having a second dataset with lower voxel size would strengthen the generalizability for hippocampal studies.

5mm resampling has been added, and the range of smoothing kernels has been extended. Results for 5mm isovoxels differ in several ways from smaller voxel sizes, showing clear disadvantages – e.g., activation in hippocampal is lost, and activation in SMC becomes diffuse, extending beyond the hand representation. 

Note that image resolution at acquisition in this study is consistent with those of other studies; 1-3mm voxels mentioned in the introduction were the resampled size used in their analysis. Examples of their image resolution at acquisition:

2.3x2.3x3.5mm with 0.5mm gap, resliced to 2mm isotropic voxels (Liu 2017) 3.75x3.75x4mm with no gap, resliced to 2mm isotropic voxels (Forcato 2016)

3.125x3.125x4mm with no gap, resliced to 3mm isotropic voxels (Malivoire 2018)

 3x3x3mm with 0.6mm gap, resampled to 3mm isotropic voxels (Danker 2016)

 3.125x3.125x4.0mm with 0.5mm gap, resampled to 2mm isotropic voxels (Qin 2014)

Regarding smoothing choice, if small regions are activated, large smoothing kernels may systematically bias or even obscure evidence of underlying activation (Friston et al., 1994). Spatial smoothing could systematically alter the localization of activation foci. This is what the investigation is showing: larger smoothing provides a larger “activation” area with smaller amplitude. See the figure 2 with the 6 to 10mm smoothing comparison. While a smaller smoothing enables to show a larger activation in a specific locus, a bigger smoothing would resemble a larger, more average and unspecific response. This does not mean we are learning more about the hippocampal response.

Figure 2 has been removed, along with Figure 6B. The effect of smoothing on activation of individual voxels is shown in a new analysis, with effects in the hippocampus showing heterogeneity in the timecourse of activation among nearby voxels. 

Related to this point of the meaning of the metric that is used in this estimation: While the task used might, as propose the author, be more robust and might display less individual variability, the link between the sensorimotor cortex and the hippocampus is not well described and can be challenging to use as a ground truth to evaluate parameters. 

Because hippocampal connectivity with SMC during these movement tasks are preferentially directed to the finger representations (Burman 2019), a well-defined area from numerous studies, this cortical link might be particularly appropriate because specific vs. nonspecific spatial effects of processing parameters may be differentiated. Hippocampal influences on all cortical areas are indirect, so there is no inherent advantage to studying a different, oft-studied connection – perhaps the contrary, in fact, since expectations about the results may be biased from previous studies. “This study specifically examined the effects of voxel size and smoothing kernel on hippocampal activation and connectivity with SMC; however, results should apply to other hippocampal connectivity studies as well, as observed effects from changes in processing parameters are consistent with previous studies.” 

In any case, this paper is not intended to promote a universal approach so much as to demonstrate that experimental findings may be overlooked when constrained by current dogma about processing hippocampal data. In the current study, the topography of its connectivity with SMC would otherwise be overlooked. 

On top of this, assuming that larger activation or connectivity (line 346) “reflect the sensitivity of the method” is not supported by the literature. Another metric related to the reproducibility of the results need to be used.

The statement here has been removed, saving the discussion of sensitivity until later.

Note that sensitivity in fMRI analyses is defined as “the probability of detecting an activation given it exists” (Hopfinger et al, 2000: A Study of Analysis Parameters That Influence the Sensitivity of Event-Related fMRI Analyses). Provided that all of the voxels show an actual effect rather than an artifact of the procedure, a method that shows larger activation or connectivity is “more sensitive” by definition. Analysis and discussion have been added to address the validity of the results, while acknowledging that replication would show how widely the methodological findings can be generalized.

Additionally, to avoid this investigation to be dataset specific and in order to generalize to other studies, this estimation could be done in relation to data characteristic. Mainly I am thinking of signal-to-noise map of individual data. The hippocampus is localized in regions with higher noise component, but this is not described for this dataset and especially not put in relation to the optimization. The idea could be to assess the reproducibility of results using a voxel size/smoothing optimization in relation to level of local noise. How does noise affect the response and reproducibility of the activation/connectivity ? Knowing the signal to noise map of an acquisition: how small can we go in voxel and smoothing to reduce noise but keep an optimal local specificity?

The signal-to-noise characteristics in a subject’s hippocampus is now characterized, but does not reliably translate to connectivity results. In the new Figure 2, the hippocampal minimal and maximal activation voxels were localized from the most representative subject from the group (as evaluated by activation volume and SMC maxima location). The elevated signal at the activation maxima fared better with smaller (3mm) voxels and little or no smoothing, whereas connectivity fared better with the 4mm voxels and 10mm smoothing. Across subjects, hippocampal minima and maxima did not consistently generate SMC connectivity. 

The new figure explains these results. Due to varied temporal characteristics in adjacent voxels, different voxel / smoothing combinations generated different eigenvariate waveforms used in PPI analysis. The combination of 4mm voxels and 10mm smoothing was optimal for demonstrating connectivity, reflecting the aggregate waveform from multiple hippocampal voxels as well as the smoothed waveform in SMC. 

Another methodological point for this study to be relevant in the field of hippocampal study is the choice of the segmentation of the hippocampus. As we see in the functional seed analysis, where we maximise the regional response strength: voxel size and smoothing do not affect the results much. Splitting the hippocampus into structural regions should also be optimized to reduce noise in the result by itself: did the author consider other, specifically individual-based segmentation and subfield segmentation? One example is the SACHA module (http://brainvisa.info).

I considered individual-based segmentation, but group analysis in SPM requires the same spatio-temporal structure for all subjects, which was accomplished by applying a normalization procedure from within SPM. The Anatomy Atlas within the WFUPickatlas can assign the probable subfield for each hippocampal region based on this normalization process. Although relevant to examine functional localization within the hippocampal formation, this is a different study.

The structural seed approach is suitable for those cases where data characteristics for the individual do not allow a prior selection of connectivity seeds. Without significant hippocampal activation or a theoretical rationale, there was no clear basis for selecting one region of the hippocampus over another, and analysis added to this study shows an individual’s activation maxima / minima does not reliably generate SMC connectivity. The functional seeds in this study resulted from an iterative procedure that effectively accounts for individual variability; significant structural seed connectivity for the group provided a region to narrow the search, the functional seed from an individual selected as the voxel within this area that generated the greatest connectivity. If we could identify this location a priori, voxel size and the smoothing kernel would matter little.

Note that if seed selection had been limited to individual minima or maxima, however, the topography of connectivity would not have been demonstrated. Connectivity depends on the temporal pattern of activity from a localized area, not the net activity of the entire region.

Last methodological point is the interpretation of the connectivity multivoxel versus ROI average comparison. A direct comparison between the two approach of connectivity might not be relevant as this choice is dependent on the question that is under investigation. Using radius around a local maximum is used from previous study / literature generalization and to account for variability and noise around a reference locus. This should not be used to represent a ROI and so cannot be compare to an average or weighted average or other time series extraction from a ROI. Not the same amount of information is represented. This choice of atlas/segmentation/time series extraction are described in connectivity analysis guideline literature. This is an example https://pubmed.ncbi.nlm.nih.gov/23583357/

The experimental question under investigation should always guide seed selection. As noted in the discussion, however, seed selection in many studies is approximate; this is the basis for the multivoxel seed analysis used here. 

Several methods to account for individual variability during seed selection may be used, yet these methods are not always effective for identifying hippocampal connectivity. Because the hippocampus carries information in its temporal pattern of activity, the local SNR and activation characteristics do not necessarily provide an accurate guide to seed placement. This provides an advantage to the structural seed approach, which does not restrict seed placement based on these uncertain factors. 

Minor comments :

Line 121 : two datasets are mentioned for 3 and 4mm, is there two independents dataset or is it the same dataset copied with two different sampling choice ? This was unclear from the text.

Clarified. This was the same dataset, resampled during the normalization step. 

Line 134: the reference of WFUPickatlas toolbox is missing and proposed only later in the text.

A link to this resource has been added at its first mention.

Line 179 : “To minimize asymmetry effects, comparisons in connectivity across voxel sizes were limited to the left hippocampus.” This is confusing because left and right analysis are proposed in the results.

Two motor tasks were studied: sequence learning and reptitive tapping. For connectivity analysis, only the left hippocampus was used. The use of 3mm voxels during connectivity analysis was limited to the sequence learning task to avoid possible interactions between hemisphere and voxel size; connectivity in this task previously shown to arise dominantly from the left side (noted now in the introduction).

Connectivity analysis was also applied to the repetitive tapping task, where the demonstration of connectivity required global bilateral analysis. For logistical reasons, this analysis only used 4mm voxels.

Fig 2 is missing brain slice numbers.

The original Figure 2 has been removed.

Fig 3 could use a bar stack code legend. The figure description talks about dash and dot and red/blue, but there is no explanation for grey, full colour or not, small or big dash, small or big or regular dots.

A legend has been added to facilitate interpretation. Bar colors and textures are designed to facilitate quick comparisons across seed locations.

Reviewer #2: This study aimed to identify optimal processing parameters of smoothing the hippocampus during functional connectivity by manipulating already acquired data. The author showed that larger voxels and smoothing kernels improve the detection with regards to hippocampal activations and connectivity. I think rhe study has incremental findings to the field and the results were obvious to the researchers especially with regards to the usage of a larger smoothing kernel. This study, however, applied multiple tests and analyses that were not needed, in my opinion, and made the attention distracted.

The original report included several subtle effects that have been removed, including figures Fig 1C, Fig 2 and Fig 6B (showing differences in hippocampal activation with different voxel sizes and extensive overlap in SMC activation).

The introduction is perfect. However, some studies are missing from being cited or not mentioned. For example, some studies investigated the effects of smoothing on resting-state functional connectivity, which is relevant to this study as the basic acquisition is the same. Also the study did not show any discussion about PPI...and why in particular it was used.

References showing smoothing effects on resting-state analysis have been added.

PPI is task-specific and directional, a point now addressed in the introduction as well as Validity and limitations. Effects can be examined related to psychological / cognitive state; hippocampal influences on mnemonic motor processes during the sequence learning task, for example, can be examined independently from hippocampal influences on non-mnemonic motor processes during repetitive tapping. 

The abstract should identify PPI.

The abstract specifies psychophysiological interactions as the means used to study connectivity; the commonly-used acronym PPI has now been added parenthetically.

I also suggest the author go throughout the whole study and emphasis that the connectivity here was effective connectivity rather than just functional connectivity (to differentiate it from resting-state connectivity)

This is now described in the introduction and again in the section on Validity and limitations.

Also, the authors wrote "This study identified optimal processing parameters" while what I can see is they just focused on smoothing.

This study identifies optimal parameters for smoothing and voxel size; the role of voxel size during connectivity analysis has now been expanded. These two parameters are most variable and controversial across published reports. 

The methods were written acceptably. However, I had to read it several times to understand the details mentioned. I strongly suggest the authors to include a demographic summarising the methods including the performed task.

A hyperlink has been added to the foundational paper (Burman, 2019), where Fig 1 illustrates one cycle from the performed task. An overview of data processing and analysis has been added as a supplementary figure.

It seems to the reader at first that the task is a pure motor task but then it seems also it has memory components.

Two motor tasks were used for analysis: a sequence learning task (with a memory component), and a repetitive tapping task (without memory). These tasks were described in the foundational article (Burman 2019), which provided the basis for the current analysis. The initial findings of this study, including differences between the two motor tasks, are now summarized in the introduction.

The voxel size of the acquisition is large. This is surprising giving the aim of focusing on the hippocampus. Also manipulating this voxel size in the preprocessing so that it goes to be isotropic or larger/smaller than what it seems make wondering about the accuracy and the meaning of this? For example, it could be much better and acceptable if the author acquired at least two sets of data with high resolution and low-resolution voxels. In this, the representations of the underlaying measured signals is not affected by the changes of the voxels at the preprocessing steps.

As noted above, the voxel size at acquisition is comparable to other studies; resampling is not done until the normalization step, and is thus unaffected by prior preprocessing steps (such as slice timing correction and realignment). Acquiring separate sets of data with high resolution and low-resolution voxels would introduce additional sources of variability that would confound the findings. 

The kernels of smoothing were selected based on what? Also I see in the tables and figures that 0 smoothings was tested while nothing in the method speaks about this.

The 0-smoothings condition has been added to the methods, and the range of smoothing kernels has been extended so as to better demonstrate which range of kernels is optimal.

The statistical tests using excel are very questionable. The author could test these using SPM or any other functional software by dealing of this dataset as different conditions or groups.

SPM can only do statistical comparisons between data when they are uniform in size and dimensions; as necessary, data were extracted for statistical analysis in Excel to overcome this limitation.

"For 3mm isovoxels, 278 voxels were identified in the left hippocampus" How about the right side? and why not mentioned?

The PPI analysis using the 3mm isovoxels involved the sequence learning task, which was limited to the left hippocampus (278 voxels); the repetitive tapping task required global bilateral PPI analysis, which was limited to 4mm analysis (78 voxels in each hemisphere). These laterality findings from the previous report are now summarized in the introduction. 

I think the discussion is acceptable. However methodological considerations of the study (not other studies) should be discussed

Methodological considerations are discussed in a new section, Validity and limitations.

---

## [Decision Letter · Decision Letter 1]

14 Jul 2021

PONE-D-20-38639R1

Topography of hippocampal connectivity with sensorimotor cortex revealed by optimizing smoothing kernel and voxel size

PLOS ONE

Dear Dr. Burman,

Thank you for submitting your manuscript to PLOS ONE. After careful consideration, we feel that it has merit but does not fully meet PLOS ONE’s publication criteria as it currently stands. Therefore, we invite you to submit a revised version of the manuscript that addresses the points raised during the review process.

The reviewers appreciated the answers to their comments. Nevertheless, after a detailed re-reading of your article, some points (detailed below) still seem to them to need clarification. For instance, it would indeed be interesting to report certain analyses at the individual level to increase the relevance of this study.

We look forward to receiving your revised manuscript.

Kind regards,

Jessica Dubois, Ph.D.

Academic Editor

PLOS ONE

Reviewers' comments:

Reviewer's Responses to Questions

**Comments to the Author**

1. If the authors have adequately addressed your comments raised in a previous round of review and you feel that this manuscript is now acceptable for publication, you may indicate that here to bypass the “Comments to the Author” section, enter your conflict of interest statement in the “Confidential to Editor” section, and submit your "Accept" recommendation.

Reviewer #1: (No Response)

Reviewer #2: (No Response)

2. Is the manuscript technically sound, and do the data support the conclusions?

Reviewer #1: Partly

Reviewer #2: Partly

3. Has the statistical analysis been performed appropriately and rigorously? 

Reviewer #1: Yes

Reviewer #2: No

4. Have the authors made all data underlying the findings in their manuscript fully available?

Reviewer #1: No

Reviewer #2: Yes

5. Is the manuscript presented in an intelligible fashion and written in standard English?

Reviewer #1: Yes

Reviewer #2: Yes

6. Review Comments to the Author

Reviewer #1: I thank the author for the thorough response to my previous comments. I appreciated the clarification of several of the comments and I would like to comment on several points that this discussion has brought up to light.

The addition of a validity and limitation section is helping to frame the use of the result from this investigation. However, I still believe such an article is difficult to apply to other datasets of potential reader as it mainly relies on specific analysis outputs, and do not provide a measure of reproducibility. This could influence reader to test all range of parameters to obtain the largest activation without questioning false negative.

Seeing that 3mm and no smoothing is more advantageous in an individual (for activation) versus lower resolution and higher smoothing for group analysis makes me wonder of the reproducibility of the results and the trend behind this optimization choice. Could a summary of the same individual analysis be provided for the dataset, in order to visual variability in the sample and not referring only to the most representative subject? Single subject analyses are also relevant for lab focusing on specific subject, precision medicine and case studies.

Additionally, or alternatively, variance map in activation result across the individuals result could help understand the change in effect between resolution and smoothing.

Can a summary be provided for half split reproducibility of group analysis, in order to strengthen the choice of resolution X smoothing effect and show a framework for other reader that would need to do this investigation to select their own optimal parameter for their specific task designs?

As an additional minor comment, the first paragraph of the validity and limitation is lacking a recall of reference.

Reviewer #2: I am happy with all the responses made by the author apart from one major comment related to the resampling of voxels sizes. The author stated that "As noted above, the voxel size at acquisition is comparable to other studies; resampling is not done until the normalisation step, and is thus unaffected by prior preprocessing steps (such as slice timing correction and realignment). Acquiring separate sets of data with high resolution and low-resolution voxels would introduce additional sources of variability that would confound the findings."

I disagree with this response since resampling or "playing" with the data set is :

- not a common practice in the field and not everyone would test it

- done on an already low resolution dataset

Therefore, the comment made by the author regarding acquiring other dataset with high or low res may add sources of variability among subjects could be true. Nonetheless, this could be overcome easily with taking into considerations these changes into a perfectly balanced statistical model and in theory would result in a better estimation than the current methods and findings of this paper.

I would not reject the paper because of this point. However, the author should clearly state this in the limitation part. Until then, I am happy to recommend accepting this paper.

I also suggest the author to re-consider the conclusion of this paper where it stated "This study shows that processing fMRI data with a larger voxel size (4mm) and smoothing kernel (8-10mm) can improve sensitivity..." This recommendation is I think overestimated given the small size of the hippocampus and the type of the acquired manipulated dataset. I hope the conclusion to be more realistic and precisely give a statement with regard to the used manipulated dataset and applied methodology. Also what was the sensitivity factor (golden reference/standard) that based on which the author reached to this conclusion.

7. PLOS authors have the option to publish the peer review history of their article (what does this mean?). If published, this will include your full peer review and any attached files.

Reviewer #1: **Yes: **Roselyne Chauvin

Reviewer #2: **Yes: **Adnan A.S. Alahmadi

---

## [Author Response · Author response to Decision Letter 1]

20 Aug 2021

Responses to reviewer comments are provided below each comment.

Reviewer #1 comment: I thank the author for the thorough response to my previous comments. I appreciated the clarification of several of the comments and I would like to comment on several points that this discussion has brought up to light. The addition of a validity and limitation section is helping to frame the use of the result from this investigation. However, I still believe such an article is difficult to apply to other datasets of potential reader as it mainly relies on specific analysis outputs, and do not provide a measure of reproducibility. This could influence reader to test all range of parameters to obtain the largest activation without questioning false negative.

Reply: There are limits to what can be done to generalize results from one study, but in the absence of a well-defined region of activation to justify seed placement, there are advantages to using the approach described in this study. I have added to the introduction a statement that succinctly summarizes the purpose of this paper: “The purpose of the methodological component of this study is not to identify optimal parameters for every case, but to demonstrate that parameters commonly used in hippocampal studies may not be optimal for connectivity studies with cortex.” 

 Informally, the methods and parameters identified can be applied to other studies. Using the optimal parameters described here, I have run PPI analyses on tasks from previous studies, finding found noteworthy results (e.g., connectivity with task-related brain areas during tasks such as Stroop, language tasks emphasizing different linguistic components, several movement tasks, and memory tasks). Data acquisition from these studies ranged from 3mm to 4mm isovoxels; after finding no discernible differences in summary figures, I decided to use 4mm isovoxels because of major differences in processing demands and disk space requirements. (Using both voxel sizes, the current study exceeded my 1TB disk space, requiring three computers to complete.) I’ve been hesitant to submit findings from these other studies until the current paper is published, due to strong opposition I’ve faced from previous reviewers over my choice of processing parameters. 

 Neuroimaging studies are not designed to exclude false negatives. A negative finding does not mean that an effect is not there, only that observed differences could statistically occur by chance (i.e., greater than a 5% probability). This explains why statistical standards used in research reports are not applied in the clinical setting (e.g., presurgical planning for brain tumor patients), where consequences of a false-negative finding are far more devastating than a false-positive. Researchers utilize those parameters likely to produce positive results; this study provides a method for evaluating connectivity throughout the hippocampus when activation is not available to guide seed placement.

Comment: Seeing that 3mm and no smoothing is more advantageous in an individual (for activation) versus lower resolution and higher smoothing for group analysis makes me wonder of the reproducibility of the results and the trend behind this optimization choice. Could a summary of the same individual analysis be provided for the dataset, in order to visualize variability in the sample and not referring only to the most representative subject? Single subject analyses are also relevant for lab focusing on specific subject, precision medicine and case studies. Additionally, or alternatively, variance map in activation result across the individuals result could help understand the change in effect between resolution and smoothing.

Reply: A summary of individual activation analysis has been added in a supplementary figure (S3 Fig). The mean magnitude of activation and standard error from all individual was plotted at the activation maximum and adjacent voxels, both for 3mm and 4mm isovoxels across both motor tasks. Before smoothing, the area of activation was comparable for both (12-16mm diameter). This activation disappeared with smoothing for 4mm isovoxels in both tasks and for 3mm isovoxels in the sequence learning task; thus, the advantage for finding activation with 3mm isovoxels depended on the analysis. Plotting beta estimates for activation across voxels at three smoothing kernels (0, 6, and 10) showed changes in activation amplitude with smoothing where adjacent voxels show sharp gradients in activation, but not where adjacent voxels responded the same. Consistent with the 12mm activation radius surrounding the maxima, this suggests a 10mm smoothing kernel does not exceed the width of a functional unit of activity in the hippocampus; instead, effects are reduced at transitional areas where the activity of nearby voxels differ sharply.

 Emphasis on activation analysis on an individual level in this study seems misplaced, however, given that A) hippocampal activation was not observed during group analysis, B) activation maxima were shown not to generate connectivity consistent with functional seeds, and C) the topography shown for connectivity would not be evident from activation maxima. Connectivity represents moment-to-moment changes in activity that were not captured by activation analysis. 

Comment: Can a summary be provided for half split reproducibility of group analysis, in order to strengthen the choice of resolution X smoothing effect and show a framework for other reader that would need to do this investigation to select their own optimal parameter for their specific task designs?

Reply: A half split reproducibility of group analysis works best for large samples, unlike the sample size from this study.

Comment: As an additional minor comment, the first paragraph of the validity and limitation is lacking a recall of reference.

Reply: A reference has been added.

Reviewer #2 comment: I am happy with all the responses made by the author apart from one major comment related to the resampling of voxels sizes. The author stated that "As noted above, the voxel size at acquisition is comparable to other studies; resampling is not done until the normalisation step, and is thus unaffected by prior preprocessing steps (such as slice timing correction and realignment). Acquiring separate sets of data with high resolution and low-resolution voxels would introduce additional sources of variability that would confound the findings."

I disagree with this response since resampling or "playing" with the data set is :

- not a common practice in the field and not everyone would test it

- done on an already low resolution dataset

Reply: Labs do not commonly resample at multiple resolutions, as they are not trying to optimize parameters -- and based on their past experience and results, they assume their parameters are fine. As detailed in my previous cover letter, most studies actually do resample at a higher resolution than what was acquired, as greater resolution during acquisition would limit brain coverage or increase TR. The current study does not undercut results from previous studies, but simply points out that findings may be overlooked with suboptimal processing parameters.

Comment: Therefore, the comment made by the author regarding acquiring other dataset with high or low res may add sources of variability among subjects could be true. Nonetheless, this could be overcome easily with taking into considerations these changes into a perfectly balanced statistical model and in theory would result in a better estimation than the current methods and findings of this paper.

I would not reject the paper because of this point. However, the author should clearly state this in the limitation part. Until then, I am happy to recommend accepting this paper.

Reply: The following statement has been added: “Hippocampal connectivity studies acquired with different resolutions and involving other cortical areas are needed to elucidate how widespread the optimal parameters from this study may be applied.”

Comment: I also suggest the author to re-consider the conclusion of this paper where it stated "This study shows that processing fMRI data with a larger voxel size (4mm) and smoothing kernel (8-10mm) can improve sensitivity..." This recommendation is I think overestimated given the small size of the hippocampus and the type of the acquired manipulated dataset. I hope the conclusion to be more realistic and precisely give a statement with regard to the used manipulated dataset and applied methodology. Also what was the sensitivity factor (golden reference/standard) that based on which the author reached to this conclusion.

Reply: The conclusion has been modified as shown below, making it clear that improved sensitivity from these processing parameters occurred here but may not occur in all studies. There is no “gold standard” in the literature for measuring sensitivity, but demonstrating topography in hippocampal connectivity with SMC required these larger smoothing kernels and voxel sizes, as smaller parameters failed to demonstrate connectivity across the breadth of the hippocampus. As such, these parameters were more sensitive in this study, since sensitivity in fMRI analyses is defined as “the probability of detecting [a property] given it exists” (Hopfinger et al, 2000: A Study of Analysis Parameters That Influence the Sensitivity of Event-Related fMRI Analyses).

New conclusion: “This study shows that processing fMRI data with a larger voxel size (4mm) and smoothing kernel (8-10mm) can, at least in some cases, improve sensitivity to hippocampal activity and its connectivity with cortical areas; optimizing these processing parameters in this study uncovered topography in hippocampal connectivity along two axes.”

---

## [Decision Letter · Decision Letter 2]

28 Sep 2021

PONE-D-20-38639R2Topography of hippocampal connectivity with sensorimotor cortex revealed by optimizing smoothing kernel and voxel sizePLOS ONE

Dear Dr. Burman,

Thank you for submitting your manuscript to PLOS ONE. After careful consideration, we feel that it has merit but does not fully meet PLOS ONE’s publication criteria as it currently stands. Therefore, we invite you to submit a revised version of the manuscript that addresses the points raised during the review process.

Reviewers are generally satisfied with your responses to their comments. However, the first reviewer had a final comment on the scope of the article. It would be appropriate to add a few sentences on this subject in the conclusion of the article. The article will not be sent back to the reviewers, I will do the final proofreading.

We look forward to receiving your revised manuscript.

Kind regards,

Jessica Dubois, Ph.D.

Academic Editor

PLOS ONE

Journal Requirements:

Reviewers' comments:

Reviewer's Responses to Questions

**Comments to the Author**

1. If the authors have adequately addressed your comments raised in a previous round of review and you feel that this manuscript is now acceptable for publication, you may indicate that here to bypass the “Comments to the Author” section, enter your conflict of interest statement in the “Confidential to Editor” section, and submit your "Accept" recommendation.

Reviewer #1: All comments have been addressed

Reviewer #2: All comments have been addressed

2. Is the manuscript technically sound, and do the data support the conclusions?

Reviewer #1: Yes

Reviewer #2: Partly

3. Has the statistical analysis been performed appropriately and rigorously? 

Reviewer #1: Yes

Reviewer #2: No

4. Have the authors made all data underlying the findings in their manuscript fully available?

Reviewer #1: No

Reviewer #2: No

5. Is the manuscript presented in an intelligible fashion and written in standard English?

Reviewer #1: Yes

Reviewer #2: Yes

6. Review Comments to the Author

Reviewer #1: I thank the author for the changes in introduction and conclusion and the addition of the supplementary figure. It gives a better frame for the reader on the aim of the article. While the sample size brings limitation to the generalization of this optimization, this article provides a rethinking frame for detecting appropriate area for connectivity analysis depending on data resolution and context. To the extent of what the data use in this investigation can offer, the case of activation-connectivity relationship is illustrated.

This article is a methodological case example. However any study using this optimization method cannot do so to the expense of a meaningful description of the variability and effect of confounds on their resulting connectivity maps and inferences. If this is clear, I have no further comment.

Reviewer #2: (No Response)

7. PLOS authors have the option to publish the peer review history of their article (what does this mean?). If published, this will include your full peer review and any attached files.

Reviewer #1: **Yes: **Roselyne Chauvin

Reviewer #2: **Yes: **Adnan A.S. Alahmadi

---

## [Author Response · Author response to Decision Letter 2]

7 Oct 2021

Reviewer #1: I thank the author for the changes in introduction and conclusion and the addition of the supplementary figure. It gives a better frame for the reader on the aim of the article. While the sample size brings limitation to the generalization of this optimization, this article provides a rethinking frame for detecting appropriate area for connectivity analysis depending on data resolution and context. To the extent of what the data use in this investigation can offer, the case of activation-connectivity relationship is illustrated.

This article is a methodological case example. However any study using this optimization method cannot do so to the expense of a meaningful description of the variability and effect of confounds on their resulting connectivity maps and inferences. If this is clear, I have no further comment.

This article may be considered a methodological case example, demonstrating that processing parameters commonly used to study activation within the hippocampus may not be optimal for identifying its influence on cortical areas. Regardless of methodology, however, a meaningful description of the variability and effect of confounds on the resulting connectivity maps and inferences should always be considered; connectivity measurements reflect the net influence of all factors affecting hippocampal activity and its connections. 

Reviewer #2: [No comments]

---

## [Editor Report · Decision Letter 3]

8 Nov 2021

Topography of hippocampal connectivity with sensorimotor cortex revealed by optimizing smoothing kernel and voxel size

PONE-D-20-38639R3

Dear Dr. Burman,

We’re pleased to inform you that your manuscript has been judged scientifically suitable for publication and will be formally accepted for publication once it meets all outstanding technical requirements.

Kind regards,

Jessica Dubois, Ph.D.

Academic Editor

PLOS ONE
---

## [Editor Report · Acceptance letter]

17 Nov 2021

PONE-D-20-38639R3 

Topography of Hippocampal Connectivity with Sensorimotor Cortex Revealed
by Optimizing Smoothing Kernel and Voxel Size 

Dear Dr. Burman:

I'm pleased to inform you that your manuscript has been deemed suitable for publication in PLOS ONE. Congratulations! Your manuscript is now with our production department. 

Kind regards, 

on behalf of

Dr. Jessica Dubois 

Academic Editor

PLOS ONE